# Structure and dynamics of the drug-bound bacterial transporter EmrE in lipid bilayers

Alexander A. Shcherbakov [1], Grant Hisao [2], Venkata S. Mandala [1], Nathan E. Thomas [2], Mohammad Soltani[3], E. A. Salter[3], James H. Davis Jr. [3], Katherine A. Henzler-Wildman [2✉] & Mei Hong [1✉]

The dimeric transporter, EmrE, effluxes polyaromatic cationic drugs in a proton-coupled manner to confer multidrug resistance in bacteria. Although the protein is known to adopt an antiparallel asymmetric topology, its high-resolution drug-bound structure is so far unknown, limiting our understanding of the molecular basis of promiscuous transport. Here we report an experimental structure of drug-bound EmrE in phospholipid bilayers, determined using $^{19}$F and $^{1}$H solid-state NMR and a fluorinated substrate, tetra(4-fluorophenyl) phosphonium (F$_4$-TPP$^+$). The drug-binding site, constrained by 214 protein-substrate distances, is dominated by aromatic residues such as W63 and Y60, but is sufficiently spacious for the tetrahedral drug to reorient at physiological temperature. F$_4$-TPP$^+$ lies closer to the proton-binding residue E14 in subunit A than in subunit B, explaining the asymmetric protonation of the protein. The structure gives insight into the molecular mechanism of multidrug recognition by EmrE and establishes the basis for future design of substrate inhibitors to combat antibiotic resistance.

[1] Department of Chemistry, Massachusetts Institute of Technology, 170 Albany Street, Cambridge, MA 02139, USA. [2] Department of Biochemistry, University of Wisconsin at Madison, Madison, WI 53706, USA. [3] Department of Chemistry, University of South Alabama, Mobile, AL 36688, USA. ✉email: henzlerwildm@wisc.edu; meihong@mit.edu

Antibiotic resistance is a rising public health crisis. Active drug efflux by multidrug resistance (MDR) transporters is of particular concern because it allows bacteria to mount rapid defense against toxic compounds. Active efflux of harmful metabolites, antiseptics, antibiotics, and toxins that either naturally occur in the environment or are produced by competing bacteria or host organisms allow bacteria to survive these challenging conditions. The broad substrate specificities of MDR transporters provide redundancy and can lead to unexpected outcomes with inhibition of individual transporters. To effectively curb this antibiotic resistance mechanism, a molecular understanding of the mechanism of substrate recognition and specificity is crucial. Here, we present atomic-level experimental distance restraints that define the location of a substrate in the transport pathway of EmrE. EmrE is a member of the Small Multidrug Resistance (SMR) transporter family in *E. coli*. In vivo, it has been implicated in pH and osmotic stress response[1], biofilm formation[2], and resistance to many quaternary cationic compounds, including the topical antiseptic acriflavine[3]. Many SMR transporters have been implicated in resistance to clinically relevant drugs in pathogenic organisms such as *mycobacterium tuberculosis*[4] and *Acinetobacter baumanii*[5]. Although EmrE has not been directly involved in resistance to clinical antibiotics by pathogenic *E. coli*, resistance is readily selected for in vitro, by mutation of only 1–3 residues[6]. Thus, structure determination of the EmrE-substrate complexes is relevant for elucidating the mechanism of action of the SMR family of transporters. In addition, EmrE is one of the smallest known proton-coupled transporters and thus serves as a model for understanding proton-coupled transport[7].

To date, the available EmrE structural models have modest resolution and lack details for understanding how multiple substrates are recognized by the protein. EmrE transports a wide array of polyaromatic cations in vitro, including ethidium, methyl viologen, acriflavine, dequalinium, and tetraphenylphosphonium[8]. Low-resolution cryo-electron microscopy (EM) maps and nuclear magnetic resonance (NMR) chemical shift changes suggest that the protein's transmembrane (TM) helices undergo large-scale reorientation to bind and transport these diverse substrates, but these data do not indicate how the substrates interact with specific residues in the protein[9,10]. Biophysical and mechanistic studies have revealed unexpected complexity in the transport process[11–17], giving evidence that EmrE may function not only as a proton-coupled antiporter, pumping toxic polyaromatic cations out of *E. coli* (Fig. 1a), but also as a proton-coupled symporter or uncoupled uniporter[11,13]. Either symport or uniport has the potential to confer *susceptibility* rather than resistance because the inward proton motive force and negative-inside membrane potential in bacteria would lead to concentrative *uptake* of toxic cations. To elucidate how EmrE interacts with and transports diverse substrates with divergent biological outcomes, high-resolution structural information of substrate-bound EmrE is essential.

The functional unit of EmrE is an antiparallel, asymmetric homodimer. Cryo-EM, X-ray crystallography, and electron paramagnetic resonance studies have established the global topology and asymmetry of the dimer. Cryo-EM maps of EmrE in 2D crystals formed in lipid bilayers[18] gave the first indication that the dimeric protein had no obvious twofold symmetry. This surprising result was later confirmed in moderate resolution (7.5 Å in-plane, 16 Å perpendicular to the membrane) 3D cryo-EM structures with and without the tight-binding substrate tetraphenylphosphonium (TPP$^+$)[19]. These 2D maps and 3D structures[9,20] also indicated that the substrates bind at the homodimer interface, but asymmetrically between the two subunits. A subsequent 3.8 Å crystal structure of the backbone of

TPP$^+$-bound EmrE[21] showed an antiparallel topology for the homodimer, which was controversial because it was the first example of a dual topology integral membrane protein. However, subsequent NMR and single-molecule fluorescence resonance energy transfer experiments on wild-type EmrE in lipid bicelles demonstrated that the asymmetric antiparallel dimer was capable of undergoing alternating-access motion (Fig. 1a)[22], and cross-linking that blocked this process also blocked transport in vivo[15]. Dual topology was further supported by mutagenesis[23,24], DEER EPR[25], and studies of the EmrE homolog Gdx[26]. The recent discovery of a second family of antiparallel homodimeric bacterial membrane proteins, the Fluc channels[27], further established dual topology. Orientational data from solid-state NMR confirmed the asymmetry of the EmrE dimer[16,28], and showed that the glutamate residue (E14)[29], which binds both protons and substrates, differs between the two subunits. Despite the well-established dual topology and asymmetry of EmrE, no atomic structure of the substrate-binding site is known.

The lack of higher-resolution structural information for EmrE is not surprising, because the protein is small and lacks large soluble domains that are usually required to obtain high-quality crystals or particle alignment for cryo-EM studies. The protein is also flexible and dynamic, as evidenced by NMR[13–16,22,28], EPR[25,30], and cryo-EM[9,20] data. These dynamics are important for multidrug recognition and transport but pose challenges for structure determination. NMR is one of the best techniques for determining the structure of small and dynamic proteins, but the conformational plasticity of the substrate-free EmrE still leads to significant line broadening in the spectra. Even with substrate-bound protein, the rate of alternating-access motion is faster than the cross-peak buildup rates in 2D correlation spectra.

Here, we report an experimental high-resolution structure of the substrate-bound EmrE using magic-angle-spinning (MAS) solid-state NMR spectroscopy. We exploit a recently discovered point mutant of EmrE, S64V-EmrE, which has the same affinity for TPP$^+$ and related substrates but slower internal dynamics and alternating-access rates[31]. Further, we use a fluorinated TPP$^+$ derivative, tetra(4-fluorophenyl) phosphonium (F$_4$-TPP$^+$) (Fig. 1a), which resembles TPP$^+$ in the transport activity, and employ a multidimensional $^1$H-$^{19}$F NMR technique[32] to measure a large number of protein–substrate distances. These distances constrain the binding-site structure. Together with $^{19}$F-detected substrate dynamics, these data provide fresh atomic insights into the mechanism of promiscuous substrate recognition and transport by EmrE.

## Results

Our structure determination of the EmrE-TPP$^+$ complex is enabled by three recent experimental advances: a long-range multiplexed $^1$H-$^{19}$F distance measurement technique[32], fluorinated TPP$^+$, and the slow-exchanging S64V-EmrE[31]. Because of the large magnetic dipole moments of $^1$H and $^{19}$F spins, $^1$H-$^{19}$F distances up to ~2 nm can now be measured using a two-dimensional rotational-echo-double-resonance (REDOR) NMR technique[32]. To utilize this technique, we synthesized F$_4$-TPP$^+$ (Supplementary Fig. 1A, B), which has the same three-dimensional structure as TPP$^+$ but slightly larger cationic charge of the central phosphorous owing to the electronegative fluorines (Supplementary Fig. 1C). To assess whether fluorination affects the protein conformation, we measured the chemical shifts of S64V-EmrE bound to F$_4$-TPP$^+$ versus TPP$^+$ under identical conditions in dimyristoylphosphatidylcholine/ di-C6-phosphatidylcholine (DMPC/DHPC) bicelles ($q = 0.33$, pH 5.8, 45 °C) using solution NMR. The amide chemical shift difference between proteins with bound F$_4$-TPP$^+$ versus TPP$^+$ (Fig. 1b) is

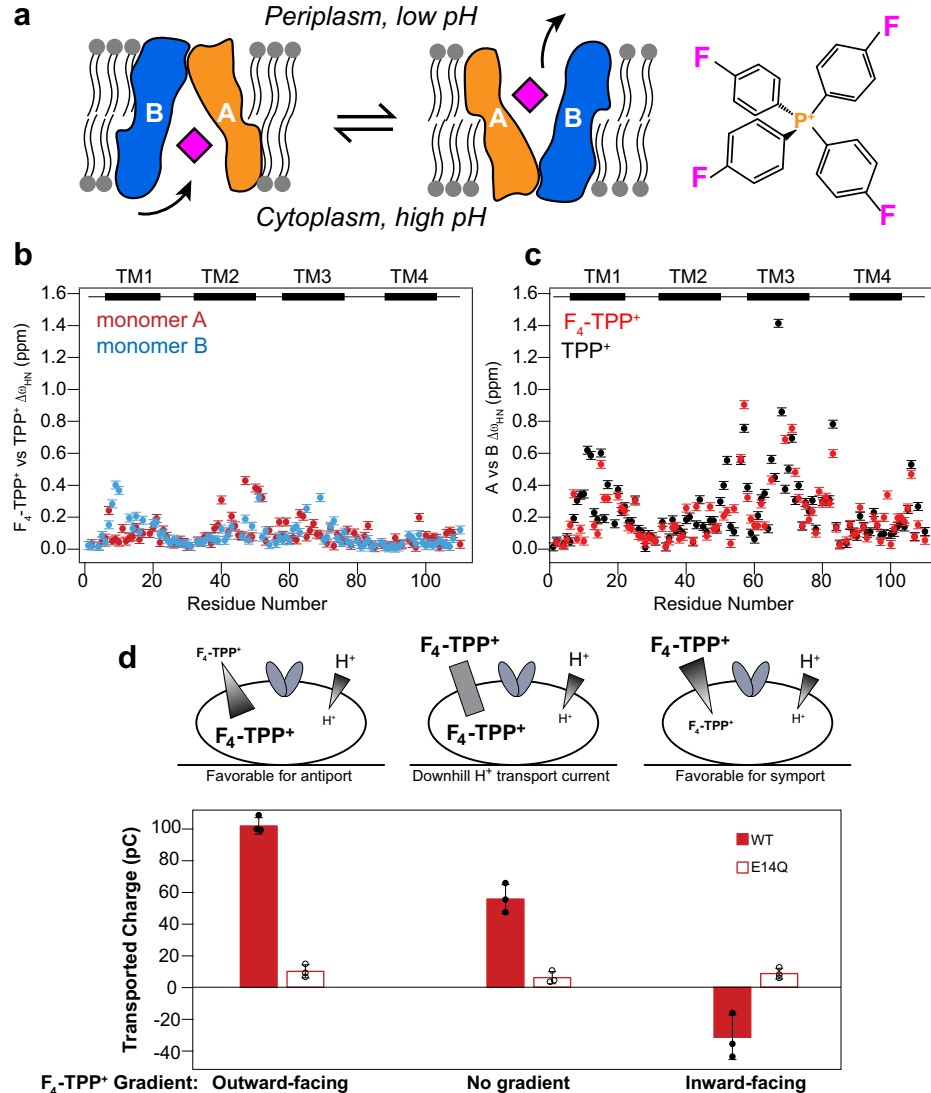

**Fig. 1 EmrE binds and transports F$_4$-TPP$^+$ in a similar fashion to TPP$^+$. a** Schematic model of the alternating-access mechanism of the asymmetric EmrE homodimer to export polyaromatic substrates out of bacterial cells. The F$_4$-TPP$^+$ structure is shown on the right. **b** Amide H$^N$ and $^{15}$N chemical shift difference between F$_4$-TPP$^+$ and TPP$^+$ bound S64V-EmrE in lipid bicelles. Red: subunit A; Blue, subunit B. Small chemical shift differences are observed, indicating that TPP$^+$ fluorination has little effect on the protein structure. Error bars are ±0.025 ppm based on spectral resolution. **c** Amide H$^N$ and $^{15}$N chemical shift difference between subunits A and B of bicelle-bound S64V-EmrE. Red: F$_4$-TPP$^+$ bound protein data; Black: TPP$^+$-bound protein data. The structural asymmetry between the two subunits is similar for the two substrates. Error bars are ±0.025 ppm based on spectral resolution. **d** Solid-supported membrane electrophysiology data of F$_4$-TPP$^+$ transport by wild-type EmrE, driven by an inward pH gradient. When the F$_4$-TPP$^+$ gradient is in the opposite direction from the pH gradient, net current increases compared with when the drug gradient is absent. When the substrate gradient is in the same direction as the pH gradient, net current decreases. Thus, F$_4$-TPP$^+$ is a canonical antiported substrate of EmrE. The E14Q mutant data serve as controls. Raw current traces and additional details are shown in Supplementary Figure 3. Error bars represent the standard error of three replicates using independently prepared sensors and the three individual data points are shown as circles.

small, and is less than the chemical shift difference between the two subunits of the dimer (Fig. 1c). The chemical shift differences between F$_4$-TPP$^+$- and TPP$^+$-bound proteins mainly localize to residues that are known from mutagenesis to interact with the substrate[30,33,34]. 2D ZZ exchange spectrum of F$_4$-TPP$^+$ bound protein shows no conformational dynamics within 200 ms (Supplementary Fig. 2), indicating that the alternating-access motion is slower for S64V-EmrE bound to F$_4$-TPP$^+$ than to TPP$^+$[31]. This slow alternating-access rate with bound F$_4$-TPP$^+$ facilitate the measurement of protein-substrate distances. But given the lack of observable alternating access, we first verified that F$_4$-TPP$^+$ is indeed transported by EmrE.

**F$_4$-TPP+ is an antiported substrate of EmrE.** We used solid-supported membrane electrophysiology to monitor liposomal transport of F$_4$-TPP$^+$ by wild-type (WT) EmrE (Fig. 1d, Supplementary Fig. 3). The assay starts with equal concentrations of both proton and substrate on either side of the liposome. Lowering the pH of the external buffer creates an inward-facing proton-concentration gradient that triggers transport, and net charge movement is recorded by the sensor. Combining this pH gradient with varying F$_4$-TPP$^+$ gradients allows for investigation of the proton/substrate coupling behavior of EmrE. In the absence of a drug gradient, protons are transported down their concentration gradient into the liposome, creating a positive

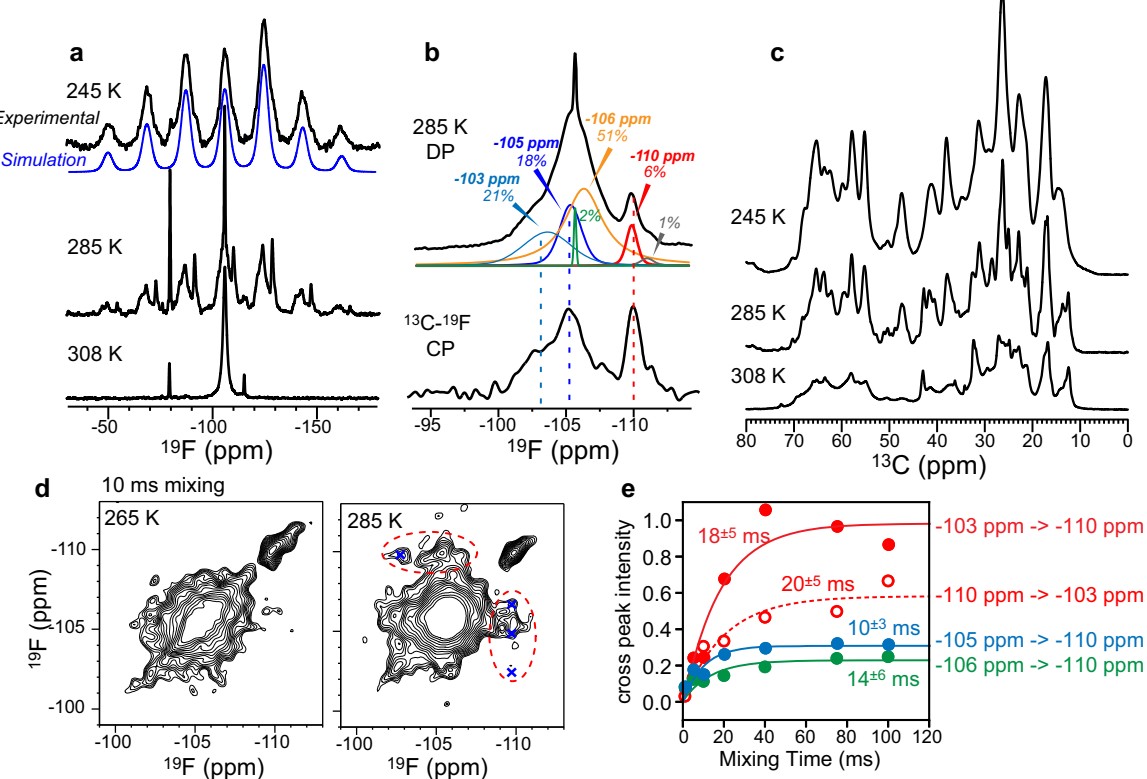

**Fig. 2 $^{19}$F NMR spectra of $F_4$-TPP$^+$ bound to S64V-EmrE in DMPC bilayers. a** Variable-temperature $^{19}$F direct-polarization (DP) spectra measured under 10.5 kHz MAS. The substrate has an isotropic $^{19}$F chemical shift of −106 ppm. The $^{19}$F linewidths and spinning sideband intensities are highly sensitive to temperature. At 245 K, $F_4$-TPP$^+$ is immobilized, as seen by the high sideband intensities, which are simulated (blue) to give the $^{19}$F CSA. In contrast, at 308 K, $F_4$-TPP$^+$ is nearly isotropically mobile. The small sharp peaks at −80 ppm and −116 ppm in the high-temperature spectrum are attributed to residual 4-fluoroiodobenzene and tris(4-fluorophenyl)phosphine from the $F_4$-TPP$^+$ synthesis. **b** $^{19}$F DP spectrum measured at 285 K under 35 kHz MAS. Spectral deconvolution gives five components, indicating that the ligand experiences a heterogeneous structural environment. $^{13}$C-$^{19}$F cross-polarization (CP) spectrum enhanced three out of the five components, indicating that these species are closest to the $^{13}$C-labeled protein. **c** Variable-temperature $^{13}$C CP MAS spectra of DMPC-bound S64V-EmrE. The spectral intensity decreases with increasing temperature, indicating that the protein becomes more dynamic at higher temperature. **d** 2D $^{19}$F-$^{19}$F correlation spectra of $F_4$-TPP$^+$ with 10 ms mixing, measured under 38 kHz MAS. Exchange peaks are detected at 285 K but not at 265 K, indicating that the exchange is owing to substrate reorientation. **e** Intensity buildup curves of cross peaks (shown as blue crosses in **d**) yield an average exchange time constant of 16 ± 2 ms.

signal for the transported charge. This signal is increased by an outward-facing $F_4$-TPP$^+$ concentration gradient (favoring anti-port). In contrast, a large inward-facing $F_4$-TPP$^+$ gradient reverses the net transport direction, indicating that protons are driven out of the liposomes against their concentration gradient. This reversal of current is indicative of coupled transport[35,36] and demonstrates that $F_4$-TPP$^+$ is antiported by EmrE. Although the timescale of transport differs between transporters, a similar reversal of current is observed for proton/guanidinium antiport by the EmrE homolog Gdx[36].

**The substrate is dynamic in the binding pocket of membrane-bound EmrE.** The four fluorine atoms of $F_4$-TPP$^+$ provide a direct probe of substrate dynamics and location with respect to the protein. The $^{19}$F NMR spectra of $F_4$-TPP$^+$ bound to S64V-EmrE in DMPC bilayers show strongly temperature-dependent spinning sideband intensities and linewidths (Fig. 2a). At a sample temperature ($T_{eff}$) of 245 K, the $^{19}$F linewidth is ~5.7 ppm, and the sideband intensity envelope fits to a rigid-limit chemical shift anisotropy (CSA) of 60.1 ppm and an asymmetry parameter of 0.8, indicating that the drug is immobilized.[37] At 285 K, most sideband intensities remain, but each peak in the sideband manifold resolves into multiple components, indicating that the fluorines experience a heterogeneous environment. At 308 K, in

the liquid-crystalline phase of the DMPC bilayer, the $^{19}$F spectrum collapses into a narrow isotropic peak at −106 ppm, indicating that the ligand undergoes nearly isotropic motion at rates faster than the $^{19}$F CSA of 34 kHz. These substrate dynamics coincide with the onset of protein dynamics, as seen in the $^{13}$C NMR spectra, which exhibit lower intensities above the membrane phase transition temperature (Fig. 2c).

To further understand the heterogeneous environment of $F_4$-TPP$^+$, we measured the $^{19}$F direct-polarization spectrum under 35 kHz MAS to detect only the isotropic peaks. The spectral lineshape is complex, and can be deconvoluted into five components (Fig. 2b). A small sharp peak (0.2 ppm linewidth) at −105.7 ppm can be attributed to free $F_4$-TPP$^+$ in solution. Three broad peaks with linewidths of 1.9–4.2 ppm are observed from −102 to −108 ppm. Most interestingly, a sharp peak with a linewidth of 1.0 ppm is observed at −110 ppm. These four components vary in intensities from 6% to 50% of the full spectral intensity, thus cannot be simply attributed to each of four fluorines. When the $^{19}$F intensity was transferred from protein carbons by cross polarization (CP)[38,39], the −103 ppm, −105 ppm, and −110 ppm peaks are preferentially enhanced, indicating that these resonances arise from fluorines that lie in close proximity to the protein carbons. The −110 ppm signal shows the largest intensity increase, indicating that this peak results from a

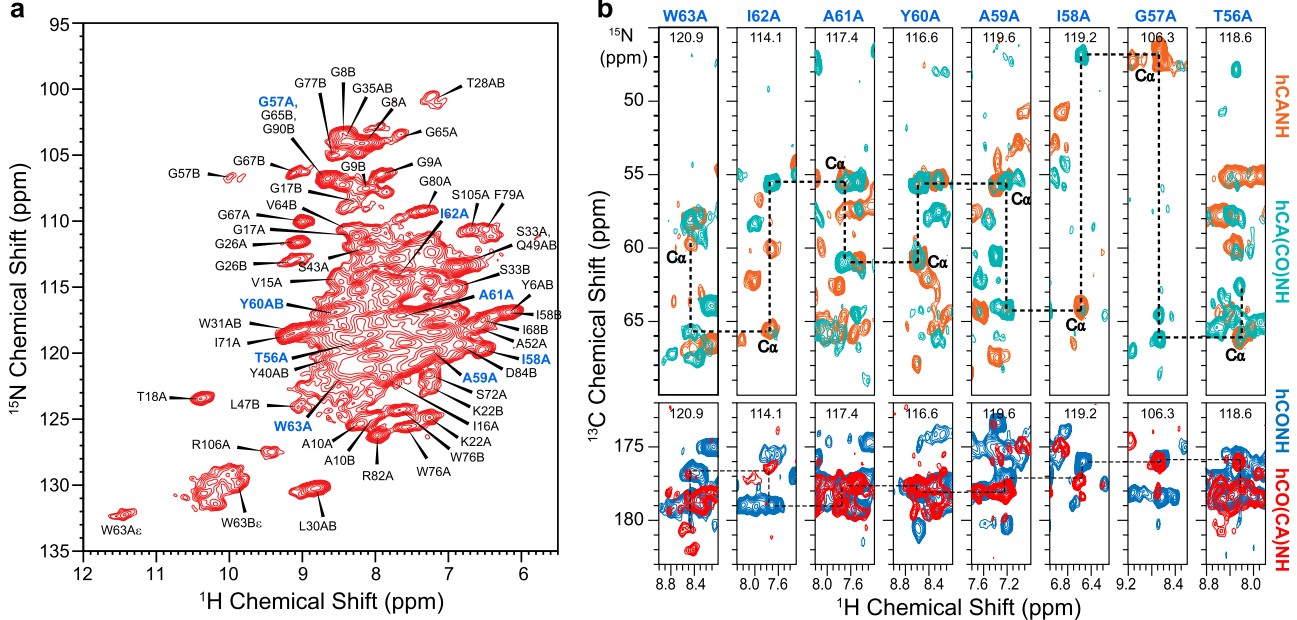

**Fig. 3 1H-detected 2D and 3D correlation NMR spectra of F₄-TPP⁺ bound S64V-EmrE in DMPC bilayers.** All spectra were measured under 55 kHz MAS on CDN-labeled protein at a sample temperature of 285 K. **a** 2D ¹H-¹⁵N correlation spectrum, showing assignment of selected resonances based on the 3D spectra. Blue assignments correspond to residues shown in **b**. **b** Representative 3D strips extracted from the four ¹H-detected spectra, showing the assignment of residues T56–W63 of subunit A. Aliphatic ¹³C chemical shifts were assigned using the hCANH (orange) and hCA(CO)NH (green) experiments, whereas CO chemical shifts were assigned using the hCO(CA)NH (red) and hCONH (blue) experiments.

fluorine that is both structurally ordered and the closest to the protein. In contrast, the −106 ppm peak is preferentially suppressed in the ¹³C-¹⁹F CP spectrum, indicating that this fluorine is the furthest from the protein. We attribute this −106 ppm peak partly to lipid-bound F₄-TPP⁺, consistent with previously reported ³¹P and ¹³C spectra of TPP⁺[40], which detected nonspecific lipid-bound ligand.

The partially resolved ¹⁹F isotropic chemical shifts allow us to probe millisecond-timescale dynamics of the substrate using a 2D ¹⁹F-¹⁹F exchange experiment (Fig. 2d). Using a fast MAS frequency of 38 kHz and a short mixing time of 10 ms, we minimized spin-diffusion effects and focused on detecting motional exchange[41]. No cross peaks are detected at 265 K, consistent with the absence of ¹⁹F-¹⁹F spin diffusion under this condition. In contrast, at 285 K, cross peaks between −110 ppm and other peaks are observed, indicating that F₄-TPP⁺ reorients on the 10 ms timescale. Cross-peak intensity buildup (Fig. 2e) indicates a time constant of 16 ± 2 ms for the exchange, indicating that F₄-TPP⁺ reorients, possibly by tetrahedral jumps, in the binding pocket with a rate of ~50 s⁻¹ at ambient temperature.

**Conformation of EmrE in DMPC bilayers.** To investigate the conformation of EmrE in lipid bilayers and to obtain the amide ¹H chemical shifts that are required for measuring protein–substrate Hᴺ-F distances, we recorded four ¹H-detected 3D MAS correlation spectra of CDN-labeled EmrE that was back-exchanged in protonated buffer. The hCANH and hCO(CA)NH spectra correlate intra-residue chemical shifts, whereas the hCA(CO)NH and hCONH spectra correlate inter-residue chemical shifts[42] (Supplementary Fig. 4). At 55 kHz MAS, the DMPC-bound protein exhibits narrow linewidths of 0.2 ppm for ¹H, 0.8 ppm for ¹⁵N, and 0.5 ppm for ¹³C, indicating high conformational homogeneity. Fig. 3 and Supplementary Fig. 5A show representative 2D strips of the 3D spectra to illustrate resonance assignment. The majority of monomer A signals show higher intensities than monomer B signals (Supplementary Fig. 6B),

indicating that monomer B in the dimer is more dynamic. This trend is reversed for TM4 residues, which show higher intensities for monomer B residues than monomer A[13]. Additional sidechain ¹³C chemical shifts were obtained from a 3D NCACX spectrum (Supplementary Fig. 5B). In total, we assigned the Hᴺ, ¹⁵N, and ¹³Cα and ¹³CO chemical shifts of 72 residues in monomer A and 54 residues in monomer B (Supplementary Table 1). The Cα and CO chemical shifts confirm that the protein is predominantly α-helical[43], with inter-helical loops at residues 25–35, 50–55, and 75–85, in good agreement with the secondary structure determined in bicelles (Supplementary Fig. 7) and with low-resolution cryo-EM and crystal structures[6,21]. The average chemical shift differences between the two subunits are small: 0.47 ppm for Cα and CO, 0.30 ppm for ¹Hᴺ, and 1.0 ppm for ¹⁵N. Among the four TM helices, TM3 displays the largest conformational asymmetry between the two subunits: for example, V64 CO, I68 Cα, and I71 Cα exhibit ¹³C chemical shift differences of 2.9 ppm, 5.4 ppm, and 3.3 ppm, respectively (Supplementary Fig. 6A). The chemical shifts show excellent agreement between bilayers and bicelles (Supplementary Fig. 8), with average ¹³C, Hᴺ, and ¹⁵N chemical shift differences of 0.23 ppm, 0.29 ppm, and 0.78 ppm, respectively, indicating that the substrate-bound EmrE conformation is similar between these two environments.

**¹H–¹⁹F distances restrain the structure of the substrate-binding pocket.** With the ¹⁵N and ¹H chemical shifts assigned, we turned to the ¹H-¹⁹F REDOR experiment[32] to measure protein–substrate distances. We detected REDOR dephasing in 2D hNH spectra, which exhibit both backbone Hᴺ signals and the sidechain indole Hε signals of the important residue W63 (Fig. 4a). Two REDOR spectra were measured at each mixing time, one without ¹⁹F pulses ($S_0$) and one with ¹⁹F pulses ($S$) to induce distance-dependent dipolar dephasing. The difference spectrum, $\Delta S$, selectively exhibits the signals of protons that are in close proximity to the fluorine atoms. Thus, the difference spectra

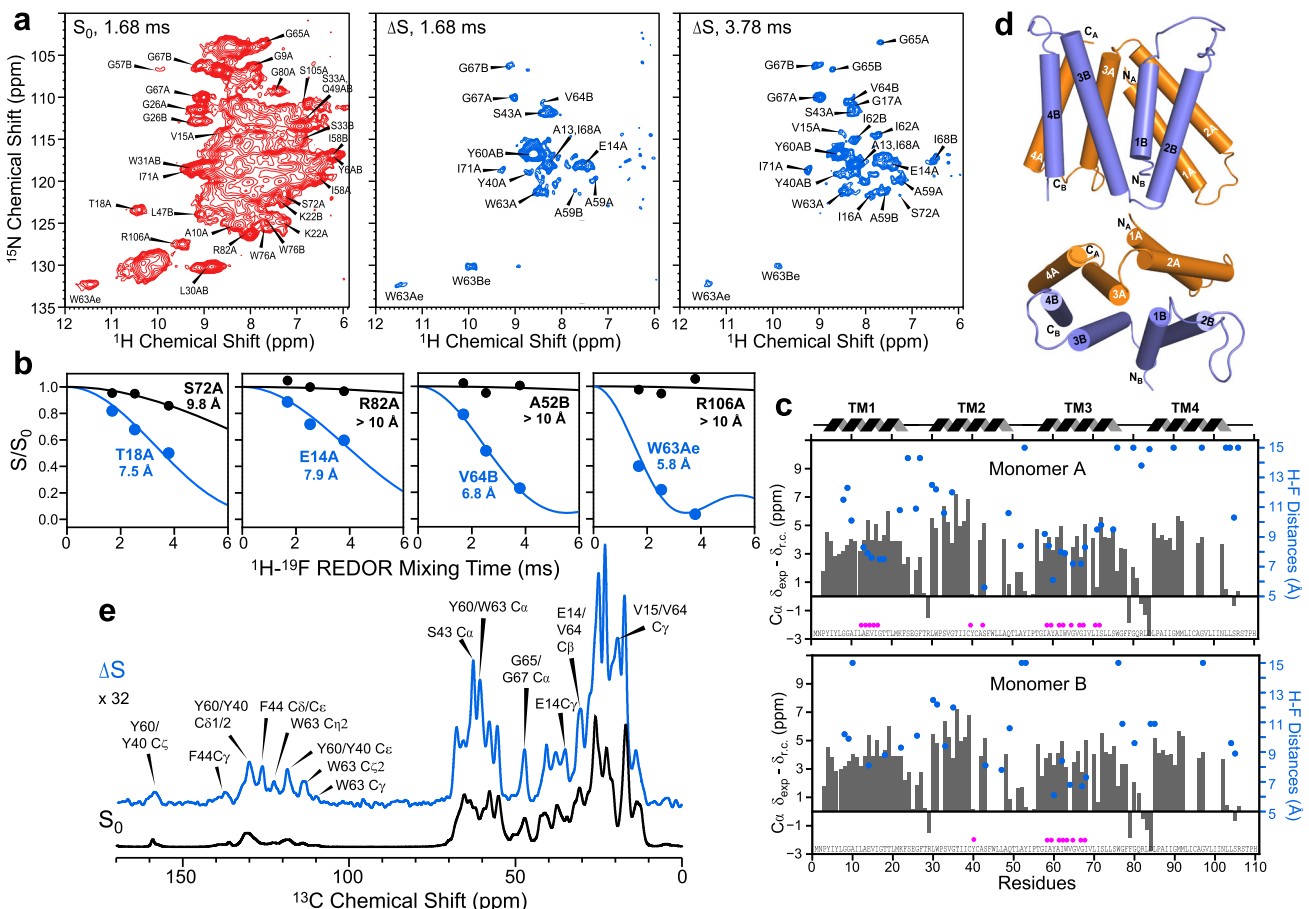

**Fig. 4 Protein–substrate distance measurement using $^1$H-$^{19}$F and $^{13}$C-$^{19}$F REDOR experiments. a** Representative $S_0$ (red) and $\Delta S$ (blue) 2D REDOR-hNH correlation spectra, measured with mixing times of 1.68 ms and 3.78 ms. Assignment is shown for selected peaks in the $S_0$ spectrum. More difference peaks are observed in the 3.78 ms $\Delta S$ spectrum than the 1.68 ms $\Delta S$ spectrum due to the detection of $H^N$ sites further away from the substrate at longer mixing times. **b** Representative $^1$H-$^{19}$F REDOR $S/S_0$ dephasing curves with best-fit simulations. Fast and slow dephasing, corresponding to short and long distances, are shown in blue and black, respectively. **c** $C\alpha$ secondary chemical shifts (gray bars) of $F_4$-TPP$^+$ bound EmrE, indicating the four TM α-helices separated by short loops. Residues whose $H^N$ atoms show difference signals in the 2D REDOR-hNH spectra are indicated by magenta circles at the bottom. Best-fit $H^N$-F distances are indicated by blue circles. Residues in TM3A, TM3B, and TM1A display short distances to $F_4$-TPP$^+$. **d** Topology of the eight TM helices in the dimeric EmrE, with monomer A helices shown in orange and monomer B helices shown in blue. **e** 1D $^{13}$C-$^{19}$F REDOR $S_0$ and $\Delta S$ spectra, coadded from spectra recorded with mixing times of 0.92, 2.0, 3.0, and 4.5 ms. The $\Delta S$ spectrum is scaled up by 32-fold with respect to the $S_0$ spectrum to better display the signals of substrate-proximal $^{13}$C sites. Note the preferential increase of aromatic $^{13}$C intensities in the 100–160 ppm region in the $\Delta S$ spectrum compared to the control $S_0$ spectrum. This is consistent with the dominance of aromatic residues at the binding site. Selected peaks are assigned based on the chemical shifts assigned from the 3D correlation spectra (Supplementary Table 1).

not only encode distance information but also simplify the assignment of substrate-proximal protons. With a REDOR mixing time of 1.68 ms, the 2D $\Delta S$ spectrum already exhibits signals from aromatic residues such as W63 and Y60 and aliphatic residues such as E14 and S43. Increasing the REDOR mixing time yielded more $\Delta S$ peaks, consistent with the detection of additional residues that are further away from the substrate. The largest number of $\Delta S$ signals results from the TM3 helix, spanning residues A59 to S72 (Fig. 4c). Difference intensities are also observed for TM1 residues A13 to T18 in subunit A and TM2 residues such as Y40 and S43. In contrast, no difference intensities are detected for residues C-terminal to the TM3 helix, indicating that the substrate-binding pocket is comprised solely of TM1, TM2, and TM3 helices.

We quantified $^1$H–$^{19}$F distances by fitting the mixing time-dependent $S/S_0$ intensity ratios (Fig. 4b): faster decays indicate shorter $H^N$-F distances. Residues such as W63Aε show rapid dipolar dephasing, indicating that they lie immediately adjacent to the substrate. The $S/S_0$ ratios decay to ~0, indicating that all

binding sites are saturated with TPP$^+$. In comparison, some residues such as R82A show minimal decay, indicating that they are far from the fluorines. Best-fit distances were obtained by minimizing the root-mean-square deviation (RMSD) between the measured and simulated $S/S_0$ ratios (Supplementary Fig. 9). Many TM1–TM3 residues show significant dipolar dephasing that is fit to distances of <10 Å (Fig. 4c, d), whereas residues in TM4, the TM1-TM2 loop and the TM3-TM4 loop display minimal dephasing and are <~10 Å from the fluorines. These distances constitute the basis for constraining the structure of the protein-substrate complex.

**Structure of the EmrE-TPP$^+$ complex in lipid bilayers: an aromatic-rich binding pocket.** We combined 214 protein–substrate H-F distances, 186 pairs of chemical shift derived $(\phi, \psi)$ torsion angles and 95 $\chi_1$ torsion angles (Table 1) to determine the structure of the EmrE-TPP$^+$ complex. Because the solution-state chemical shifts are very similar to the solid-state values and are more extensively assigned, we used the larger set of

**Table 1 Solid-state NMR and structure refinement statistics for $F_4$-TPP$^+$ complexed S64V-EmrE structure in lipid bilayers.**

| | Monomer A | Monomer B |
|---|---|---|
| **NMR distance and dihedral constraints** | | |
| Dipolar couplings | 42 | 30 |
| Distance constraints | 120 | 94 |
| Total number of dihedral-angle restraints | | |
| $\phi$ | 99 | 86 |
| $\psi$ | 99 | 87 |
| $\chi_1$ | 53 | 42 |
| **Structure refinement statistics** | | |
| Violations (mean ± s.d.) | | |
| Distance constraints (Å) | 0.008 ± 0.064 | 0.019 ± 0.114 |
| Max. distance-constraint violation (Å) | 0.96 | 1.48 |
| $\phi$ Dihedral-angle constraints (°) | 0.160 ± 1.486 | 0.126 ± 0.995 |
| $\psi$ Dihedral-angle constraints (°) | 0.217 ± 2.143 | 0.383 ± 2.421 |
| Max. $\phi$ dihedral-angle violation (°) | 22.8 | 13.9 |
| Max. $\psi$ dihedral-angle violation (°) | 31.4 | 31.5 |
| Average pairwise r.m.s.d (Å)[a] | | |
| Protein heavy atom | 2.12 ± 0.23 | |
| Protein backbone | 1.61 ± 0.19 | |
| Ligand heavy | 1.35 ± 0.35 | |
| Ligand center[b] | 0.76 ± 0.32 | |

[a]Pairwise r.m.s.d. was calculated among 10 lowest-violation structures between the two independent MD runs after the refinement had equilibrated.
[b]Ligand center is defined as phosphorus and its four directly bonded carbon atoms of $F_4$-TPP$^+$.

torsion angles obtained solution NMR in the final structure calculation. The use of solid-state NMR chemical shifts did not make any noticeable difference. The structure calculation consisted of two stages: docking of $F_4$-TPP$^+$ into previous molecular dynamics (MD) simulated apo structural models in dimethyl sulfoxide (DMSO), followed by all-atom refinement of the docked protein–ligand complexes in explicit DMPC bilayers. The docking stage used as input the measured $^1$H-$^{19}$F distance restraints with fourfold ambiguity (Supplementary Table 2), while the MD refinement stage used as input structurally assigned $^1$H-$^{19}$F distance restraints (Supplementary Table 3) together with the protein torsion angles. Two apo structural models, biased to the low-resolution crystal structure, were used for docking[44,45]. The substrate clustered to a single position in one apo protein model[44] (Supplementary Fig. 10A) but diverged to four positions in the second model[45] (Supplementary Fig. 10B). For the latter, only one of these four positions lies at the dimer interface. Thus, we removed the outcome of the second model from further analysis. In the uniquely docked model, TPP$^+$ is surrounded by TM1–TM3 helices of subunit A and TM3 of subunit B. Among the 20 lowest-energy docked structures, the phosphorus and its four directly bonded carbons in TPP$^+$, which represent the center of the molecule, have a mean RMSD of 1.6 Å, whereas the protein shows an all-atom RMSD of 0.6 ± 0.1 Å. Importantly, docking allowed the assignment of the phenylene Hζ atoms, replaced by fluorine here, that dephase each protein H$^N$ (see Methods and Supplementary Table 3). With the $^1$H-$^{19}$F pairs thus assigned and the overall ligand position constrained, we then refined the protein structure in DMPC bilayers (Supplementary Fig. 10C) under the constraints of the measured $^1$H–$^{19}$F distances and ($\phi$, $\psi$, $\chi_1$) torsion angles.

We used two lowest-violation HADDOCK models to carry out two independent MD runs. The resulting two structural ensembles (Supplementary Fig 10D) are each well clustered and show only modest differences from each other. The protein

heavy-atom RMSD was 2.12 ± 0.23 Å for the run 1 ensemble and 1.98 ± 0.40 Å for the run 2 ensemble, whereas the RMSD between the two lowest-violation structures from each ensemble was 1.59 Å (Supplementary Fig. 10F, G). The structural differences between the two runs are manifested more in subunit B than in subunit A, with an RMSD of 1.58 Å for monomer B and only 1.13 Å for monomer A. This observation is consistent with the fact that monomer B is more dynamic than monomer A (Supplementary Fig. 6B). Excluding the loops, the largest deviations between the two ensembles are found at C-terminal ends of TM3A and TM3B (Supplementary Fig. 10D), consistent with the fact that the chemical shift asymmetry is the most pronounced for the C-terminal end of TM3 (Supplementary Fig. 6A). TPP$^+$ has the same orientation between the two ensembles (Supplementary Fig. 10F) and has a small RMSD of 0.61 Å for the center atoms. Given the overall structural similarity between the two lowest-energy ensembles, we chose 10 conformers from the two runs with the lowest violations with the experimental distance restraints to constitute the final NMR structure ensemble (Supplementary Fig. 10H).

The NMR structural ensemble of the EmrE $F_4$-TPP$^+$ complex shows TM1–TM3 residues to interact with the substrate while the two TM4 helices associate to stabilize the dimer (Fig. 5a, Supplementary Fig. 10D, H). This architecture is in good agreement with the low-resolution cryo-EM and X-ray data[21,46]. In each monomer, E14 in TM1 and Y40 in TM2 approach the substrate from one side, while Y60 and W63 of TM3 approach the substrate from another side at an angle of ~100° from the E14/Y40 pair (Fig. 5b). Among these four residues, Y40 is the furthest away from the substrate (Table 2). The relative proximities of these aromatic and polar residues to $F_4$-TPP$^+$ are in good agreement with biochemical data that W63 and Y60 are essential for substrate binding and transport, whereas Y40 regulates substrate specificity[7]. Between the A and B subunits, the two E14 sidechains are approximately colinear and lie on two opposite sides of TPP$^+$. However, the E14 displacement from the substrate is asymmetric. The distances from the phosphorous to the two E14 Cδ atoms, averaged over the 10 structures, are 5.6 Å to monomer A and 7.5 Å to monomer B. The four phenylene Hζ corners of the substrate are also asymmetrically positioned from E14: the nearest Hζ lies 4.6 Å away from E14A Cδ, whereas the nearest Hζ lies 6.5 Å from E14B Cδ. These structural features suggest that monomer A provides more stabilization energy to the substrate. This is consistent with the weaker intensities of monomer B peaks compared with monomer A, suggesting that monomer B is more dynamic. Importantly, one of the four phenylene Hζ atoms, designated as F13 (Fig. 5b), is held by a cage of four functional residues: W63A, W63B, Y60A and E14B, with distances of 5.8 Å, 5.7 Å, 6.9 Å, and 6.5 Å to W63A Nε, W63B Nε, Y60A Oζ, and E14B Cδ, respectively (Table 2). Thus, the F13-bearing phenylene ring of TPP$^+$ experiences multiple weak π-π, CH-π, and electrostatic interactions with the protein, making this fluorine most likely responsible for the narrow −110 ppm peak in the $^{19}$F spectrum (Fig. 2b). This binding-site geometry indicates that multivalent aromatic and polar interactions play the dominant role for TPP$^+$ binding to EmrE. The aromatic-rich nature of the binding pocket is further evidenced by $^{13}$C-$^{19}$F REDOR spectra (Fig. 4e), which show difference intensities at the $^{13}$C chemical shifts of residues such as W63, Y60, F44, E14, G65, G67, S43, and V64.

The NMR structure of the EmrE-TPP$^+$ complex puts TPP$^+$ at a similar location as the previous MD model of drug-bound EmrE[44] (Fig. 5c, Supplementary Fig. 10I), but differs in terms of the substrate orientation and the positions and orientations of the protein residues. In the NMR structure, the two W63 indoles are

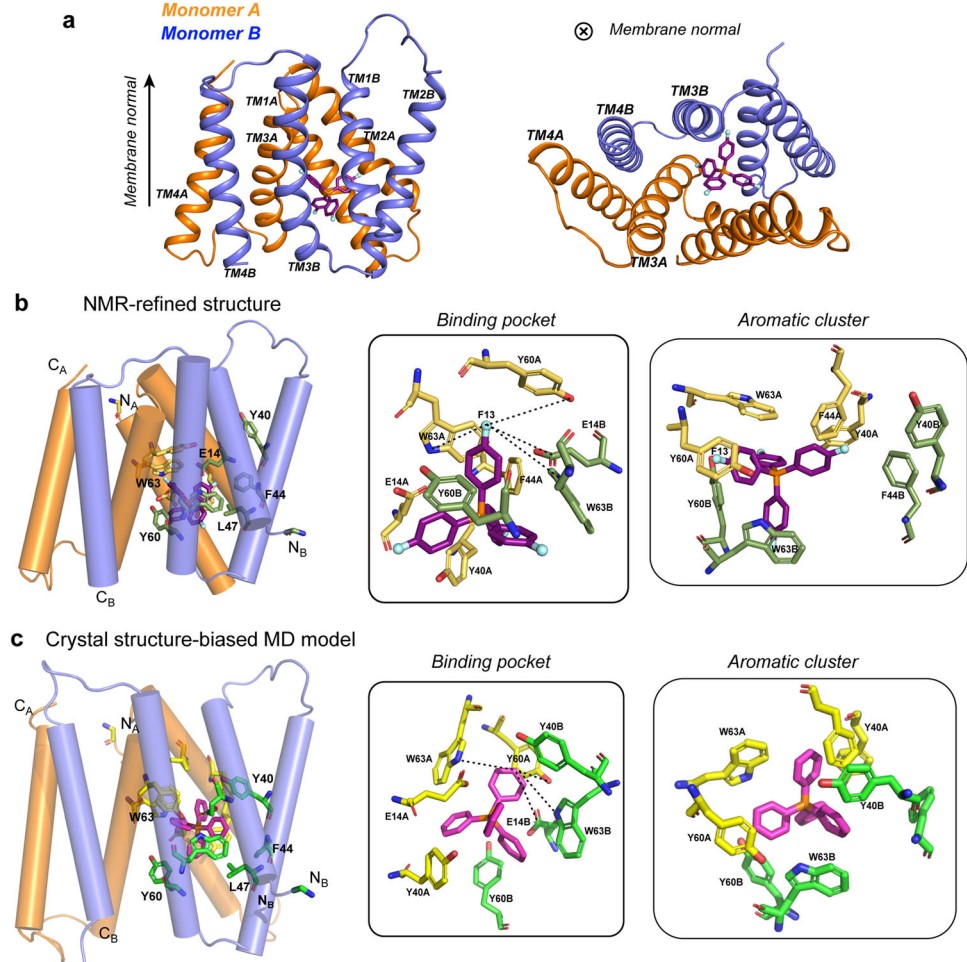

**Fig. 5 Experimentally determined structural model of the EmrE-TPP$^+$ complex in DMPC bilayers at low pH. a**. Side view (left) and bottom view (right) of the substrate-EmrE complex. The ligand (purple) lies closer to monomer A (orange) than monomer B (blue). **b** Distance-constrained NMR structure model of the drug-binding site. Key residues, including E14, Y40, Y60, and W63, surround the substrate (middle). One of the four phenylene Hζ atoms, marked as F13, is tightly coordinated by residues from both monomer A (yellow) and monomer B (green). The right panel shows a bottom view of all aromatic residues (W63, Y60, F44, and Y40) surrounding the substrate. Monomer A residues are colored in yellow while monomer B residues are colored in green. **c** Crystal structure biased MD-simulated structure model of TPP-bound EmrE. The substrate position and orientation relative to the binding-site residues differ from those in the experimental NMR structure. Monomer A residues are shown in yellow and monomer B residues are shown in green.

**Table 2 EmrE to F$_4$-TPP$^+$ distances extracted from the NMR-refined structural ensemble.**

|  | Monomer A | Monomer B |
|---|---|---|
| P—E14 Cδ | 5.6 ± 0.3 Å | 7.5 ± 1.0 Å |
| P—Y40 Oζ | 6.8 ± 0.5 Å | 16.7 ± 0.3 Å |
| P—Y60 Oζ | 9.8 ± 0.7 Å | 5.9 ± 0.4 Å |
| P—W63 Nε | 6.0 ± 0.4 Å | 5.6 ± 0.3 Å |
| Min. F [a]—E14 Cδ | 4.6 ± 0.5 Å | 6.5 ± 0.7 Å |
| Min. F [a]—Y40 Oζ | 6.2 ± 0.6 Å | 12.2 ± 0.4 Å |
| Min. F [a]—Y60 Oζ | 6.9 ± 0.4 Å | 5.6 ± 0.5 Å |
| Min. F [a]—W63 Nε | 5.8 ± 0.3 Å | 5.7 ± 0.4 Å |
| F13 [b]—E14 Cδ | 8.7 ± 0.6 Å | 6.5 ± 0.7 Å |
| F13 [b]—Y60 Oζ | 6.9 ± 0.4 Å | 5.6 ± 0.5 Å |
| F13 [b]—W63 Nε | 5.8 ± 0.3 Å | 5.7 ± 0.3 Å |

The average distances and standard deviations are from the ensemble of 10 lowest-violation structures in the final 170 ns of the two MD trajectories.
[a]Distances of the nearest fluorine to protein atoms.
[b]Distances from the F13 atom, which resides in an aromatic box formed by W63 and Y60, to protein atoms.

roughly perpendicular to each other, with the W63A indole plane at an angle of ~50˚ with respect to the nearest, F13-bearing, TPP$^+$ phenylene plane. In comparison, the MD model puts the two W63 indole rings roughly parallel to each other, and both are perpendicular to the closest TPP$^+$ ring[44]. In the MD model, the two E14 residues approach the substrate at different angles and displacement compared with those of the NMR structure. The residues in the NMR structure model are more loosely packed compared with the MD model. For example, the E14B-Y60A-W63B triad show distances of 4.0–7.0 Å among the Cδ, Oζ, and Nε atoms in the NMR structure (Fig. 5b), whereas the corresponding distances in the MD structural model are much shorter, 3.2–3.8 Å, suggesting hydrogen bonding. On the other side of TPP$^+$, the equivalent triad of E14A-Y60B-W63A is similarly loose, with inter-atomic distances of 3.7–6.4 Å.

## Discussion

The results shown here provide the first extensive experimental definition of the geometry of the substrate-binding pocket of EmrE. The large number of protein–substrate $^1$H-$^{19}$F distances (Supplementary Fig. 10E)[32], measured in bilayer-bound EmrE,

indicates the relative proximities of residues at the binding pocket. The inequivalent substrate position between the two subunits[42] gives insight into the asymmetric protonation of the two E14 residues[13,14]. In the NMR structure, E14A Cδ in monomer A is closer to the TPP$^+$ phosphorus and the nearest fluorine than E14B Cδ in monomer B (Table 2), suggesting that E14A is negatively charged and should experience favorable electrostatic attraction with the positively charged F$_4$-TPP$^+$. The inequivalent positions of E14 with respect to the substrate explain the distinct chemical shifts of E14A and E14B, with E14A exhibiting larger Cα, Cβ, and Cγ chemical shifts than E14B (Supplementary Table 1). A previous NMR study of E14-labeled EmrE bound to ethidium bromide reported similar aliphatic chemical shift differences as seen here[29], but in addition, measured the carboxyl chemical shifts at a low temperature of 200 K. E14A and E14B were found to have Cδ chemical shifts of 180.9 ppm and 178.3 ppm, respectively. Since protonated carboxyl groups have smaller isotropic chemical shifts on average than deprotonated ones[47], these Cδ chemical shifts indicate that E14B is protonated while E14A is deprotonated in the presence of ethidium. Therefore, this previous result is fully consistent with our current assignment of E14A to be negatively charged and in closer proximity to the positively charged TPP$^+$. In addition, solution NMR pH and TPP$^+$ titrations with bicelle-bound EmrE also suggested asymmetric binding of TPP$^+$: only E14B remained titratable, with a pK$_a$ of 6.8, implying that TPP$^+$ is closer to E14A, preventing its protonation[13]. These data, taken together, indicate that TPP$^+$, as well as other EmrE ligands, bind the dimer asymmetrically, closer to subunit A than to subunit B,

The current data also report the first observation of substrate dynamics in the binding pocket, which is coupled to protein dynamics. These dynamics are manifested by the $^{19}$F NMR spectra of the substrate at physiological temperature and the temperature-dependent $^{13}$C spectra of the protein (Fig. 2). It is also hinted by the subtle TM3 helix orientational difference between the two MD runs (Supplementary Fig. 10D). How does the structure of the protein–substrate complex solved at low temperature explain these motions at high temperature, and how do these motions relate to promiscuous substrate binding and transport? The current structure shows that the binding pocket is composed of many aromatic and polar residues, which engage in multivalent interactions with the substrate that are not easily perturbed by a single mutation at residue 64. At the same time, the binding pocket is spacious: most inter-residue distances are longer than the hydrogen-bond length, and the protein–drug distances are also sufficiently long to allow drug reorientation. This spacious and multivalent binding pocket explains the similar binding affinities of TPP$^+$ for the mutant and wild-type EmrE, and is also consistent with the ability of the protein to bind multiple drugs promiscuously. However, binding does not equal transport. Efficient translocation of the drug requires coordinated motion of the protein between the outward-facing and inward-facing conformations. Although S64V-EmrE binds substrates with nearly identical, sub-micromolar, affinity as wild-type EmrE, it has a slower transport rate and eightfold slower alternating-access rate than the WT protein[31]. This slower transport rate implies a reduced ability of the mutant to undergo coordinated conformational changes. We observed less helical chemical shifts for V64 in monomer B than in monomer A, suggesting helix disorder in TM3 of monomer B[9,46]. The fact that this TM3 disorder is observed in the more dynamic monomer (Supplementary Fig. 6) suggests that the local motion of monomer B might regulate the ability of the dimer to undergo conformational interconversion, which is required for drug efflux. Future comparison of the mutant with the WT structure and dynamics will be

required to determine whether increased dynamics of monomer B facilitates or impedes the alternating-access motion.

Among different members of the tetrahedral ligands, higher binding affinity is correlated with slower transport[10]. When the same ligand binds different mutant proteins, then the anti-correlation between binding affinity and protein dynamics weakens[31]. These data suggest that the ligand geometry and protein structure both affect the binding-site structure and alternating-access rate, but in a partly independent manner. Future studies to determine how the EmrE binding-site structure changes with the ligand, pH, and protein mutation, will be informative to define the conformational landscape of this promiscuous transporter.

The contribution of MDR transporters to bacterial virulence and antibiotic resistance has led to significant interest in developing efflux pump inhibitors. The goal is to block transport activity in order to reduce bacterial virulence, restore antibiotic efficacy, and provide tools to understand the complex toxin efflux network in bacteria[48–50]. These inhibitors often resemble substrates and compete for substrate binding or prevent the protein from undergoing the conformational changes that are required for transport[51]. The structure presented here provides an initial guide for structure-based design of EmrE inhibitors to probe EmrE function within the E. coli MDR efflux network in vivo. As a model system, EmrE has provided rich insight into the complexity of proton-coupled drug transport. Biophysical studies have revealed its ability to perform different types of coupled transport that would lead to either resistance or susceptibility in vivo[11,13]. Mutagenesis of EmrE and other SMR homologs demonstrate the ease with which SMR transporters may be converted between these two phenotypes and confirm that a single transporter can confer resistance to some substrates and susceptibility to others[6,52]. Application of the approach used here to additional substrates will provide a foundation for understanding the multidrug poly-specificity of EmrE and how different substrates can interact with EmrE to trigger opposing biological outcomes of resistance or susceptibility.

## Methods

**Synthesis of tetra(4-fluorophenyl) phosphonium iodide**. Into a 50 ml heavy-wall pressure vessel with a polytetrafluoroethylene internal-thread cap with a magnetic stir bar, 4-fluoroiodobenzene (1.4 g, 1.0 equiv), tris(4-fluorophenyl) phosphine (2.0 g, 1.0 equiv), Pd(OAc)$_2$ (0.021 g, 1.5 mol%), and mixed xylenes (15 mL) were added. The tube was flushed with nitrogen, capped, and the reaction mixture stirred at 140 °C for 2 hours. The product, tetra(4-fluorophenyl) phosphonium iodide, precipitates during the course of the reaction. Once cooled, the product was isolated by filtration, washed with small portions of fresh xylenes, and air-dried. The pure tetra(4-fluorophenyl) phosphonium iodide product was isolated as a pale ivory solid (3.2 g, 95% yield).

**S64V-EmrE expression and purification**. S64V-EmrE was expressed and purified following published protocol[31], using the same procedure as for WT EmrE[53]. In brief, for $^{13}$C,$^{15}$N-labeled S64V-EmrE, the protein was expressed using media containing 2.5 g/L U-$^{13}$C glucose, 1 g/L $^{15}$NH$_4$Cl, 0.5 g/L $^{13}$C,$^{15}$N-labled ISOGRO (Millipore-Sigma). $^2$H,$^{13}$C,$^{15}$N (CDN) S64V-EmrE was expressed in $^2$H$_2$O media containing 2.5 g/L U-$^2$H,$^{13}$C glucose, 1 g/L $^{15}$NH$_4$Cl, 0.5 g/L $^2$H,$^{13}$C,$^{15}$N-labled ISOGRO. $^2$H,$^{15}$N (DN)-labeled S64V-EmrE was expressed in $^2$H$_2$O media containing 1 g/L $^{15}$NH$_4$Cl and 0.5 g/L $^2$H,$^{13}$C,$^{15}$N-labled ISOGRO. Lysis and purification were performed using Ni-NTA affinity column followed by thrombin cleavage of the His-tag and size exclusion chromatography on a S200 column in buffer containing 50 mM MES, 20 mM NaCl, 10 mM decyl-maltoside, 5 mM BME, pH 5.8[10,53].

**Solid-supported membrane-based electrophysiology experiments**. WT EmrE and E14Q EmrE was expressed and purified similar to S64V-EmrE[13]. To minimize solution exchange artifacts, the buffers used for size exclusion chromatography, reconstitution, and electrophysiology steps had the same salt composition: 50 mM MES, 50 mM MOPS, 50 mM bicine, 100 mM NaCl, and 2 mM MgCl$_2$. Buffer pH values were carefully adjusted using only NaOH to ensure that internal and external Cl$^-$ concentrations were identical for all measurements. Protein was reconstituted into POPC liposomes at a 1:400 protomer: lipid molar ratio, and

detergent was removed with Amberlite XAD-2. Liposomes were collected, aliquoted, and flash frozen. Immediately prior to measurements, liposomes were thawed, diluted twofold with pH 7.3 buffer, and briefly sonicated.

All electrophysiology measurements were recorded and analyzed using a Surf2er N1 solid-supported membrane-based electrophysiology (SSME) instrument from Nanion Technologies. Prior to measurements, sensors were equilibrated on the instrument with multiple washes with pH 7.30 buffer containing 0.5 μM $F_4$-$TPP^+$ while recording currents. Washes were performed until successive washes produced no observable current. Transport was initiated by perfusion of pH 7.00 buffer containing 10 μM $F_4$-$TPP^+$ to simultaneously set inward-facing proton and drug gradients. Transport currents were recorded during 1.5 s of perfusion of the external buffer and integrated to obtain transported charge. After these measurements, sensors were washed with pH 7.30 buffer containing 10 μM $F_4$-$TPP^+$ while recording currents. Washes were again performed until successive washes produced no observable current. (0.5 μM $F_4$-$TPP^+$ for outward-facing gradient, 10 μM $F_4$-$TPP^+$ for inward-facing or no gradient. Transport was initiated by perfusion of pH 7.00 buffer containing 0.5 or 10 μM $F_4$-$TPP^+$ to simultaneously set inward-facing drug and/or proton gradients. Reported values are the average of replicates on three different sensors, and error bars are the standard error of the mean.

**Reconstitution and preparation of solid-state NMR samples**. $^{13}$C,$^{15}$N-labeled S64V-EmrE was reconstituted into DMPC (Avanti Polar Lipids) liposomes at a EmrE monomer to lipid molar ratio (P: L) of 1: 50 or 1: 25. DMPC was resuspended in 50 mM MES, 20 mM NaCl, pH 5.8 buffer at 20 mg/mL. The lipid mixture was incubated at 45 °C for 1 h to hydrate, then bath-sonicated for 1 min before addition of 0.5% octyl-glucoside followed by 30 s bath sonication. The lipid mixture was incubated at 45 °C for an addition 15 min before mixing with purified S64V-EmrE solution. After 20 min room temperature (RT) incubation, Amberlite (Supelco) was added (3 × 30 mg Amberlite per mg total detergent) to remove the detergent. The amberlite was removed after 16–24 hours by simple filtration. Liposomes were collected by ultracentrifugation (165,000 × g, 6 °C, 2 h.) and resuspended in a small volume (~20 mg/mL lipid conc.) of buffer. To ensure complete detergent removal, the sample was dialyzed against 1 L of the same buffer (50 mM MES, 20 mM NaCl, pH 5.8) with buffer change every 24 hours over a 72 hour period. The sample was then incubated with excess solid $F_4$-TPP + + at RT with end-to-end mixing for at least 16 hours. Excess $F_4$-TPP + + was removed using microcentrifugation (8000 × g, 5 min). Proteoliposomes were then pelleted at 100,000 × g, 4 °C, 2 h in an ultracentrifuge. A similar method was used to prepare the CDN-S64V-EmrE sample, except that the protein was reconstituted into DMPC-$d_{54}$ liposomes at a P: L of 1: 25. Proteoliposomes were dried to ~40% hydration by mass in a desiccator. Samples were centrifuged into 3.2, 1.9, and 1.3 mm MAS rotors. Three 1.9 mm rotors were packed: (1) a CDN-EmrE sample containing 3.6 mg protein in 16.0 mg proteoliposomes, (2) a CN-labeled EmrE sample (P: L = 1: 25) containing 3.4 mg protein in 14.9 mg proteoliposomes, and (3) a CN-labeled EmrE sample (P: L = 1: 50) containing 1.9 mg protein in 15.0 mg proteoliposomes. A 1.3 mm MAS rotor was packed with 0.9 mg CDN-EmrE in 3.9 mg proteoliposomes. The 3.2 mm Revolution NMR rotor was packed with 5 mg CN-EmrE in 39 mg proteoliposomes (P: L = 1: 50).

**Reconstitution and preparation of solution NMR samples**. All solution NMR samples were reconstituted into DMPC/DHPC isotropic bicelles (q = 3) at a 75:1 lipid to EmrE monomer ratio. The reconstitution was performed similarly to the solid-state NMR sample up to the point where liposomes were pelleted at (165,000 × g, 6 °C, 2 h). Once pelleted, the sample was resuspended in buffer containing threefold higher concentration of 1,2-dihexanoyl-sn-glycero-3-phosphocholine (DHPC-6). Samples were then subjected to three cycles of freeze-thaw. The pH of the final samples was adjusted to 5.8 at 45 °C using a Hamilton biotrode microelectrode. The sample was then incubated with excess solid $F_4$-TPP + + at 45 °C overnight. Excess $F_4$-TPP + + was removed using microcentrifugation prior to transferring solution to NMR tubes.

**Solid-state NMR experiments**. All MAS NMR experiments were conducted at 600, 700, and 800 MHz Bruker NMR spectrometers. $^1$H-$^{19}$F REDOR distance measurements were conducted under 38 kHz MAS at an effective sample temperature ($T_{eff}$) of 285 K on a Bruker Avance III HD 600 MHz (14.1 T) spectrometer at MIT using a 1.9 mm HFX probe. $^1$H-detected 3D correlation experiments were conducted on 1.3 mm HCN probes under 55 kHz MAS between 280 and 285 K on the 600 MHz spectrometer and an Avance NEO 700 MHz (16.5 T) spectrometer at Bruker Biospin (Billerica, MA). $^{13}$C-detected 3D correlation experiments were conducted under 14 kHz MAS using a 3.2 mm BlackFox HCN probe on an 800 MHz spectrometer.[54] The effective sample temperatures were estimated from the water $^1$H chemical shifts using the equation $T_{eff}$ $(K) = 96.9 \times (7.83 - \delta_{H_2O})$ where $\delta_{H_2O}$ is the measured water $^1$H chemical shift.[55] The corresponding thermocouple-reported set temperature ($T_{set}$) is given in Supplementary Table 4. There is no chemical shift difference between fast and slow MAS, and the samples were maintained at similar temperatures by choosing appropriate bearing temperatures. Thus, the protein conformation is unchanged by fast MAS compared with slow MAS.

Pulse sequences for the $^1$H-detected experiments and $^{19}$F solid-state NMR experiments are shown in Supplementary Fig. 4, whereas detailed experimental parameters are given in Supplementary Table 4. In general, N-C correlation experiments used $^{SPECIFIC}$CP for polarization transfer[56]. $^{13}$C-$^{13}$C correlation was achieved using the CORD spin-diffusion sequence[57] under slow MAS (14 kHz) and the DREAM sequence[58] for one-bond $^{13}$C–$^{13}$C transfer under fast MAS (55 kHz). High-power $^1$H decoupling used either continuous wave or TPPM[59] schemes, and low-power $^1$H decoupling was performed using the WALTZ-16 scheme.[60] Proton-detected MAS NMR experiments employed MISSISPPI to suppress the water $^1$H signal.[61] Four $^1$H-detected 3D correlation experiments were used to assign the $^1$H, $^{15}$N, and $^{13}$C chemical shifts of bilayer-bound S64V-EmrE. The hCANH and hCO (CA)NH experiments allow intra-residue assignment, whereas the hCA(CO)NH and hCONH experiment allow inter-residue assignment.

$^{19}$F chemical shifts were externally referenced to the −122.1 ppm signal of 5F-tryptophan on the $CF_3Cl$ scale and $^{15}$N chemical shifts were externally referenced to the $^{15}$N peak of N-acetylvaline at 122.0 ppm on the liquid ammonia scale. $^1$H and $^{13}$C chemical shifts were internally referenced to match the DSS-referenced chemical shifts of the solution-state $^1$H and $^{13}$C values. The solid-state 2D $^{13}$C-$^{13}$C CORD spectrum was calibrated by referencing the T28AB Cβ peak to 70.3 ppm. For the hNH, hCANH, hCONH, and hCA(CO)NH spectra, we chose G67A as the reference signal, setting the $^1$H chemical shift to 9.0 ppm, $^{13}$Cα to 47.1 ppm, the V66 $^{13}$CO to 178.1 ppm, and V66Cα to 67.1 ppm (Supplementary Table 1). The hCO(CA)NH spectrum was similarly referenced to solution-state chemical shifts. However we noticed temperature-induced perturbations between redundant $^{13}$CO shifts in the hCONH and hCO(CA)NH spectra. As a result, we calculated the average perturbation in the hCO(CA)NH spectrum relative to the hCONH spectrum for 10 $^{13}$CO shifts, and applied a +0.4 ppm correction to the $^{13}$C dimension of the hCO(CA)NH spectrum.

**Solution NMR experiments**. TROSY-selected ZZ exchange[62] spectra were collected on an 800 MHz Varian VNMRS DD spectrometer equipped with a 5 mm cold probe ($^1$H/$^{13}$C/$^{15}$N) using VnmrJ 4.0. The VT setpoint was set at 45 °C and data were collected with 200 ms mixing for ~5 days, yielding no discernable exchange cross peaks. Data were processed using NMRPipe[63] and NMRFAM-Sparky[64] was used to analyze spectrum.

**Solid-state NMR spectral analysis and distance extraction**. 1D and 2D MAS NMR spectra were processed in the Bruker Topspin software package. 3D correlation spectra were added in the frequency domain using a Python script that made use of NMRGlue and NumPy Python packages.[65,66] Chemical shift assignment and plotting of 3D spectra were performed in NMRFAM-Sparky[67]. Comparisons of solid-state and solution NMR chemical shifts and monomer A and B chemical shifts were computed in Python and plotted with Matplotlib.[68] Protein backbone torsion angles were predicted from measured chemical shifts using the TALOS-N software[43], excluding all $^1$H chemical shifts and applying a deuterium isotope correction to the Cα and Cβ chemical shifts.

$H^N$-$^{19}$F distance restraints were extracted from least square fit of the experimental REDOR data by numerically simulated curves.[32,69] In brief, peak volumes in the 2D $^1$H–$^{19}$F REDOR-hNH $S_0$ and S spectra were integrated to obtain the intensity ratios $S/S_0$ for all mixing times. We then simulated the two-spin REDOR dephasing curves for distances of 3.0–15.0 Å in 0.1 Å increments using the SIMPSON software package[70]. These numerical simulations included the magnitude (δ) and asymmetry (η) of $^{19}$F CSA, but left the tensor orientations unknown (i.e., Euler angles of 0, 0, 0). Finite-pulse effects were explicitly encoded in the NMR parameters. RF inhomogeneity was accounted for by simulating for pulse flip angles of 180° to 145° in 5° increments, weighted by a half-Gaussian function centered at 180°.[32,69] The REPULSION168 scheme with 32 gamma angles was used for powder averaging[71]. The best-fit $^1$H–$^{19}$F distance was extracted by minimizing the RMSD between the simulated and measured $S/S_0$ values. The uncertainty in the best-fit distance was set by an RMSD threshold of 0.2, as this was the maximum scatter observed for sites that do not dephase; distances below this RMSD value were considered significant (Supplementary Fig. 9). In cases where little or no dephasing was observed, we set the distance upper uncertainty to 40 Å, which is approximately the longest possible distance in the dimer. For residues whose signals overlap in the 2D hNH spectrum, the lower-limit distance uncertainty was increased.

**Solution NMR assignment spectra**. Solution chemical shift assignments were assigned using the following suite of three-dimensional NMR spectra with non-uniform sampling (NUS): TROSY-HNCA and TROSY-HNCACB spectra were recorded on a 900 MHz Bruker Avance III HD equipped with a 5 mm triple resonance cryoprobe ($^1$H/$^{13}$C/$^{15}$N) running Topspin NMR 3.5. Temperature was set to 45 °C. The TROSY-HNCA[72–74] (Bruker trhncaetgp2h3d) was acquired with 1024 complex points in the direct dimension ($^1$H) and 604 non-uniformly sampled complex points in the indirect dimensions (max increments 36 ($^{15}$N) and 48 ($^{13}$C)) for 35% sampling. Sixty-four scans were acquired per increment with a 2 s delay. Spectral widths were 16.34 ppm centered at 4.58 ppm ($^1$H), 31.3 ppm centered at 116.5 ppm ($^{15}$N) and 29.5 ppm centered at 55.6 ppm ($^{13}$C). The TROSY-HNCACB[72–74] (Bruker trhncacbetgp2h3d) was acquired with 1024 complex points

acquired in the direct dimension ($^1$H) and 740 non-uniformly sampled complex points in the indirect dimensions (max increments 36 ($^{15}$N) and 64 ($^{13}$C)) for 32% sampling. In all, 128 scans were acquired per increment with a 2 s delay. Spectral widths were the same as the HNCA except for $^{13}$C, which was 63.1 ppm centered at 43.6 ppm.

The TROSY-HN(CO)CA, TROSY-HNCO (BioPack ghnco_trosy_3DA), and TROSY-HN(CA)CO[75–77] spectra were collected on a 600 MHz Varian VNMRS DD console equipped with a 5 mm cold probe ($^1$H/$^{13}$C/$^{15}$N) using VnmrJ 4.0. The TROSY-HNCO was acquired with 1024 complex points in the direct dimension ($^1$H) and 900 non-uniformly sampled complex points in the indirect dimensions (max increments 48 ($^{15}$N) and 48 ($^{13}$C)) for 39% sampling. 32 scans were acquired per increment with a 2 s delay. Spectral widths were 20.03 ppm centered at 4.58 ppm ($^1$H), 35.4 ppm centered at 117.8 ppm ($^{15}$N) and 11.9 ppm centered at 177.4 ppm ($^{13}$C). The TROSY-HN(CA)CO was acquired with 1024 complex points in the direct dimension ($^1$H) and 380 non-uniformly sampled complex points in the indirect dimensions (max increments 32 ($^{15}$N) and 34 ($^{13}$C)) for 38% sampling. 144 scans were acquired per increment with a 2 s delay. Spectral widths were the same as for the HNCO. The TROSY-HN(CO)CA was acquired with 1024 complex points in the direct dimension ($^1$H) and 598 non-uniformly sampled complex points in the indirect dimensions (max increments 36 ($^{15}$N) and 48 ($^{13}$C)) for 35% sampling. 64 scans were acquired per increment with a 2 s delay. Spectral widths were the same as for the HNCO except for $^{13}$C, which was 29.8 ppm centered at 55.9 ppm.

All data were processed using NMRPipe[63] and SMILE[78] for NUS reconstruction. Spectral analysis and assignments were performed using CcpNmr Analysis[79].

**Electrostatic surface calculation.** The electrostatic surface of the EmrE substrates TPP$^+$ and F$_4$-TPP$^+$ were calculated assuming S$_4$ symmetry. Gaussian16 optimization was used with B3LYP/6-31 G(d) initially, and then re-optimized using B3LYP/6-311 + G(d,p). Single point B3LYP/6-311 + G(d,p) calculations were performed using Spartan'08 to generate the elstat figures.

**Structure calculation of F$_4$-TPP$^+$-bound EmrE.** Structure calculation of the TPP$^+$–EmrE complex consists of two stages. The first stage is docking of F$_4$-TPP$^+$ into the apo protein structure, to determine the location and orientation of the drug and to structurally assign which fluorine dephases each protein H$^N$. The second stage is MD simulation in explicit lipid bilayers to equilibrate and refine the structure of the protein–drug complex. The measured $^1$H-$^{19}$F distances served as the input for the docking step. Two MD-simulated apo EmrE structure models biased to the low-resolution crystal structure were used as the starting protein structure[21,44,45]. Both E14 residues are protonated, and a S64V mutation with the lowest-energy rotamer was introduced. The coordinate of F$_4$-TPP$^+$ was generated by replacing the *para*-hydrogens of TPP$^+$ [44] with fluorines. To assign the two sets of protein chemical shifts to monomers A and B, we used the V64 Cα chemical shift. A resolved Val Cα signal is observed at 64.1 ppm, which is upfield from the more ideal α-helical Cα chemical shift (~66 ppm) of all other valines. In the apo EmrE structural model, TM3 of monomer A is a relatively ideal α-helix whereas TM3 of monomer B has a significant kink around V64. Thus, we assigned the less-helical 64.1 ppm V64 Cα chemical shift to monomer B and the more ideal helical chemical shifts to monomer A. The N- and C-termini of the protein were set as charged, and the "active" residue list, from which the HADDOCK Ambiguous Interaction Restraints (AIRs) were generated, was set to a minimal subset of residues that are known to be involved in binding based on biochemical data[80] and the current REDOR data. $^1$H–$^{19}$F distance restraints with REDOR RMSD values below 0.2 (Supplementary Fig. 9) were used as unambiguous constraints, for which energy penalties were always enforced. Docking was performed in DMSO and started with 1000 structures, from which 200 lowest energy structures were refined. These 200 structures were aligned and analyzed in Pymol with an integrated Python-Pymol script for reporting the protein and ligand RMSD's.

Between the two MD models of the apo EmrE, only the Karplus model[44] resulted in the drug clustered to a single position. Thus, we used the two lowest-violation structures from this docked ensemble to resolve the ambiguity of which $^{19}$F atom(s) dephase which $^1$H atoms of the protein. We read the PDB coordinates into dataframe structures using an in-house written Python script that employed the Biopandas package[81]. We calculated the distances between all fluorine atoms and each protein H$^N$ to identify the nearest fluorine. The distances to the three other fluorine atoms were checked against the shortest distance; if any of the three other fluorines had a distance within 1.5 Å of the nearest F, then this fluorine atom was also flagged. In total, we assigned each protein H$^N$ to one of three categories: (1) those H$^N$ atoms with a single nearest F, a positive constraint; (2) those H$^N$ atoms with two similarly proximal (within 1.5 Å of each other) F atoms, or two positive constraints, and (3) those H$^N$ atoms that are far from all fluorine atoms, or four "negative" constraints. In case (2), we increased both upper-bound distance uncertainty by 2 Å, to account for the fact that the REDOR fitting assumed a single-distance two-spin model, so that the individual distance in the three-spin situation is longer than the two-spin fit. This structure-based assignment algorithm converted the ambiguous HADDOCK distance restraints to a list of pairwise distance restraints to be input into the GROMACS simulation. In total, from the

72 dipolar coupling measurements, we obtained 214 unique distance restraints for all-atom MD simulations.

To refine the protein structure in the bound complex, we conducted MD simulations on the converged HADDOCK ensemble. Two independent MD runs were initiated from the two lowest-violation HADDOCK models to ensure the consistency of the structure refinement. We inserted the docked EmrE-TPP$^+$ complexes into explicit hydrated DMPC bilayers using the CHARMM-GUI[82] membrane builder tool[83]. The DMPC bilayer contains 100 lower leaflet lipids and 104 upper leaflet lipids, and hydrated on both faces with a TIP3P water layer of ~2.5 nm in thickness[84]. A total of six chloride ions and four sodium ions were included in the system to match the experimental 20 mM NaCl condition. The complex was aligned to the membrane normal using the OPM web-service.[85] The ligand force fields were parameterized from the ligand coordinates. MD simulation was conducted in GROMACS[86] using the NMRbox virtual servers[87]. The simulation was conducted at 310 K, and CHARMM36 force fields[88,89], including the WYF parameter for cation-pi interactions, were used. Backbone ($\phi$, $\psi$) angles and sidechain $\chi_1$ torsion angles from solution NMR chemical shifts were applied as constraints with an angle uncertainty of ±20°. The protein-ligand H-F distance restraints (Supplementary Table 3) were applied with the piecewise linear restoring force in GROMACS[90]. Simulation started with a 5000-step energy minimization with position and dihedral restraints on the protein backbone, sidechain, and lipid atoms. The position and dihedral restraints were progressively weakened and removed over 1.875 ns of equilibration. The production stage involved 400 ns in 2 fs steps to equilibrate the structure. GROMACS periodic boundary condition commands were used to remove jumps across the box boundary, and the MDAnalysis Python package[91] was used to align each successive MD step to the initial state for calculating RMSDs (Supplementary Fig. 10C) and to keep the protein position immobilized throughout the trajectory. Final structures were subjected to a similar 5000-step energy minimization to remove improper bond angles. Two ensembles from the two MD runs were created taking 18 time points between 230 and 400 ns. The models from both trajectories were scored based on the original fourfold ambiguous set of distance restraints. In the 36 structures from the two MD runs, the average number of violations was 7.4 ± 1.9 (min, max = 3, 10), the average violation magnitude is 1.8 ± 0.4 Å (min, max = 1.2 Å, 3.2 Å), and the total violations ("violation score") is 12.3 ± 1.3 Å (min, max = 9.5 Å, 14.4 Å). The final reported ensemble consisted of the 10 lowest-violation conformers from the two runs; all 10 structures in the final ensemble came from run 1.

**Reporting summary.** Further information on research design is available in the Nature Research Reporting Summary linked to this article.

## Data availability
Solid-state NMR chemical shifts and distance restraints have been deposited in the Biological Magnetic Resonance Bank with ID number 50411. The structural coordinates have been deposited in the Protein Data Bank with the accession code 7JK8.

## Code availability
Python codes for $^1$H-$^{19}$F REDOR analysis, structurally based H-F pair assignment, and GROMACS simulations are available upon request to meihong@mit.edu.

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

## Acknowledgements

This work is supported by NIH grants GM095839 to K.A. H-W, GM066976 to M.H., and the MIT School of Science Sloan Fund to A.A.S. and M.H. The study made use of NMR spectrometers at the National Magnetic Resonance Facility at Madison, supported by NIH grant P41 GM103399 and P41 RR002301; the Center for Magnetic Resonance, supported by P41 GM132079; and the NMRbox, supported by P41 GM111135. Equipment was purchased with funds from the University of Wisconsin-Madison, NIH (S10RR02781, S10RR08438, S10RR023438, S10RR025062, S10RR029220), the NSF (DMB-8415048, OIA-9977486, BIR-9214394), and the USDA. This study also made use of the Alabama Supercomputing Center. We thank Jochem Struppe for help in setting up two of the $H^N$-detected 3D experiments, Albert Donkoh for assistance with probe repairs, Peyton Spreacker for S64V-EmrE sample preparation, and Marco Tonelli for help with the measurement of the solution NMR assignment experiments.

## Author contributions

K.A.H.-W. and M.H. designed the experiments. A.A.S. conducted all-solid-state NMR experiments, data analysis, and structure calculation; V.S.M. contributed to the solid-state NMR experiments, resonance assignment, and structure calculation. G.H. purified the protein, prepared the membrane samples, and conducted the solution NMR experiments; G.H. and K.A.H.-W. analyzed the solution NMR data; J.H.D. devised the $F_4$-$TPP^+$ design and synthetic approach; M.S. carried out $F_4$-$TPP^+$ synthesis, purification, and characterization; E.A.S. performed the electrostatic surface calculations; N.T. performed and analyzed the liposomal transport assays. All authors discussed the results and wrote the paper.

## Competing interests

The authors declare no competing interests.
