## [Transparent Peer Review File · Nature Communications]

REVIEWER COMMENTS

Reviewer #1 (Remarks to the Author):

This manuscript focuses primarily on the NMR structure of EmrE in a lipid bilayer that has a bound drug. This NMR work is solid but the details of computational work and approach could be improved.

General Comments:

1. Why choose DMPC for NMR work?: Since this protein is a bacterial transporter in *E. coli*, why use a non-native lipid, DMPC. PE would potentially be a more realistic choice as bacterial membranes are roughly 70-80% PE. Some discussion on this and comparison to past NMR work in various lipids (done by the corresponding author) should be clearly stated.

2. Are the MD Simulations long enough?: The authors have some detail of the stability of the structure (or lack of) in the supporting materials (Figure S9). The authors claim that Figure S9A is RMSF but this is not typically how RMSF is reported. Usually root-mean-squared fluctuations are reported based on a per residue basis (showing residue fluctuations about a STABLE average). I suspect this is the root-mean-squared deviation (RMSD). Either way, the structure has NOT stabilized. There is still a visible drift in increasing RMSD for the last 100ns. The authors must continue these simulations longer until there is at least 100ns of stable RMSD.

3. No discussion of NMR vs. MD structure differences: There lacks a discussion on why MD shows a more compact binding pocket compared to the NMR structure. There are other significant deviations from NMR. What if the authors started a second simulation imposing a structure similar to NMR? Would the structure be stable or revert to the current MD simulations based on the low-resolution x-ray structure? Currently this paper focuses strongly on NMR and lacks any discussion of the simulation work in the Discussion section and any insight in why there are differences.

4. Proper citation of force fields: The authors only cite the protein force field, when lipids and water are used. The CHARMM36 lipid force field should be cited as well as TIP3P.

Reviewer #2 (Remarks to the Author):

See attached file for more clarity

Reviewer #3 (Remarks to the Author):

When it was first described by Shimon Schuldiner, EmrE has been considered as the ideal model system for secondary transporters and multidrug efflux due to its small size and apparent simplicity. Ironically, this relatively small membrane protein with just four transmembrane helices turned out to be a complex experimental challenge. It forms an antiparallel homodimer and was the first dual topology transporter described. Extensive efforts to obtain a high-resolution structure - including NMR, Xray and cryo-EM - failed so far but at least all data emerged into a consensus model. However, the lack of real high-resolution data still prevents deriving a good mechanistic model. One key problem has been the unfavorable intrinsic dynamics of EmrE which might have prevented progress in structure determination so far.

In this paper, Hong and Henzler-Wildman present the first high resolution structure of EmrE. The work presents excellent and exciting progress in the field and was only possible due to the combination of four experimental tricks/methods: The structure determination is based on solid-state NMR, which

enables to work directly in native-like bilayers. Especially noteworthy is the fact that a large number of valuable long-range contacts could be obtained through the use of ^1H - ^{19}F REDOR experiments. For this purpose, the high affinity substrate TPP was fluorinated. Partial assignment was achieved by ^1H and ^{13}C detected 3D experiments. However, most importantly for this work has been the introduction of the S64V mutation, which alters the protein dynamics in such a way, that a great spectral resolution could be achieved. All previous solid-state NMR studies on the EmrE wild type suffered from line broadening due to an unfavorable motional regime.

In my opinion, this work is a great step forward. Data quality is excellent and data interpretation is sound, but I have some minor comments / suggestions / questions which might help improving the paper:

- For me it is still puzzling why/how a homodimer in complex with a symmetric substrate assumes an asymmetric structure and even shows 'asymmetric dynamics' as highlighted in the manuscript with A being in stronger complex with the substrate compared to B. What determines the different behavior of A and B?

- E14 plays an important role in the asymmetry argument and different protonation states in the substrate bound state could certainly explain the observations. Is it correct that only CD of E14A has been assigned (Tab. S1)? How about the different distances between TPP and E14A-CD and E14B-CD discussed in the paper? Are these experimental results or conclusions from the structure calculation?

- Substrate dynamics, heterogeneity, TPP-E14 interaction: It might be helpful to refer to Ong et al. JACS 2013 (incl. SI) in which dynamics and heterogeneity of TPP and MTP within the EmrE pocket has been addressed by ^{31}P -MAS NMR and in which ^{13}C -TPP has been used for probing interactions with E14 by DNP. Overall, these results seem to fit with the data presented here. Regarding the ^{19}F line shape analysis in Fig. 2A/B/C: Could some of the spectral components arise from non-specifically bound TPP populations as shown by Ong et al.?

- Structure calculations: How unambiguous is the structure calculation? The reason for the question is that prior structural knowledge is needed for the substrate docking step and for assigning the F-H restraints. Are there any controls which could be done? Also, in contrast to NOE or PDSO restraints, REDOR distances are precise measurements. What determined the upper and lower limits for the structure calculation? How important are the ϕ/ψ angles used here and is it a problem that not all residues have been assigned? Could the discrepancy between A and B (72 vs. 54 residues assigned) affect the structure calculation in terms of asymmetry?

Reviewer #4 (Remarks to the Author):

Multidrug resistance (MDR) is one of the greatest threats to human health. In contrast to other transporters, MDR transporters have a rather complicated mechanism due to their broad substrate specificity. EmrE is a well-studied model protein for MDR. There is evidence that EmrE is able to perform antiport, symport, and uniport under different conditions, which further enhances the complexity of its transport mechanism. This is a rather new field as transporters have been believed to be strictly coupled; By theory, antiport and symport contradict each other. For these reasons, a deeper insight into the transport mechanism of EmrE is of broad interest to the scientific community.

Structural data on EmrE is already available and helped to understand its functional asymmetry and dual topology. But the authors report the first high-resolution structure of drug-bound EmrE, which allows for important conclusions about drug-transporter interactions on the side chain level. In addition, the authors' report includes the first observation of substrate dynamics at the binding pocket. This and other mechanistic insights also help developing inhibitors to face MDR.

The authors used novel approaches to counter the issues responsible for the lack of high-resolution structural data on MDR transporters: (1) They developed a mutant (S64V-EmrE) with similar drug-binding properties, but slower conformational dynamics. (2) They applied a newly developed long-range ^1H - ^{19}F distance NMR technique. (3) And they also found a drug (F4-TPP+) with reduced

conformational dynamics.

To validate that F4-TPP⁺ is indeed a transported EmrE substrate, the authors applied SSM-based electrophysiology – a rather novel approach for the measurement of transport activity in membrane vesicles or proteoliposomes.

The authors used pH jumps to activate EmrE, which is a difficult approach due to possible solution exchange artifacts. The authors were aware of that, designed a well-suited solution exchange protocol, and used proper control experiments: The same measurements using the transport deficient E14Q EmrE mutant did not generate significant signals on the SSM, proving that the currents detected for WT-EmrE are due to proton-coupled F4-TPP⁺ transport by EmrE.

Another difficulty is the establishment of the chemical gradient for F4-TPP⁺ during the functional assay. The description in the method section indicates that the sensors were equilibrated with the respective F4-TPP⁺ concentration prior to the measurement. From that description, it is not clear if they equilibrated by solution exchange using the measurement device or during sample preparation and before sonication of the sample. When F4-TPP⁺ was equilibrated during sonication, the concentrations inside and outside the vesicles should perfectly equilibrate. But when equilibration was performed during solution exchange, the concentrations inside the vesicles may not equilibrate to the exact concentrations of the buffer due to the lipid barrier and rise in membrane potential when F4-TPP⁺ crosses that barrier. The membrane potential serves as a counterforce which may prevent further F4-TPP⁺ equilibration, even when longer incubation times are used.

However, the analysis used by the authors shows that EmrE transport activity is enhanced, when the direction of the F4-TPP⁺ gradient is in the opposite direction than the pH gradient. This makes perfect sense for an antiport mechanism. The conclusion is also perfectly valid, even when the exact concentrations inside the vesicles are not known.

For proton and F4-TPP⁺ gradients in opposite directions, a large positive charge is translocated, showing that more protons are imported than F4-TPP⁺ molecules are exported per transport cycle. When only the proton gradient serves as a driving force, the translocated charge decreases, rendering one step in F4-TPP⁺ translocation as the rate-limiting step; And showing that the authors really observe F4-TPP⁺ translocation. However, when both cationic gradients are inward-facing: Why does the translocated charge turn negative, when the transporter is activated by an increase in external proton concentration? To my understanding, this is only possible, when both gradients are established at the time to drive transport, e.g. a simultaneous F4-TPP⁺ and pH jump; or when the pH gradient is established before activation of EmrE via F4-TPP⁺ concentration jump; In these experimental settings, the result would indicate that the inward-facing F4-TPP⁺ gradient drives efflux of protons, even against the proton gradient. But in the methods section, the authors state that F4-TPP⁺ concentration is equilibrated before the measurement which is triggered by pH jump. The increase of proton concentration on the external side can only trigger proton influx; The translocated charge has to be positive and the minimum charge translocation observed (for zero transport activity) should be zero. Further clarification is required here.

Taken together, the understanding of the SSM-based electrophysiology assay would be enhanced when the following points are considered: 1. More details about the experimental settings should be given, e.g. details about pre-loading the vesicles with F4-TPP⁺, incubation times between measurements, and the time points of when the pH and F4-TPP⁺ concentrations are changed at the sensor. Are all three conditions shown measured on different sensors which have been equilibrated with the given F4-TPP⁺ concentration? Or did the authors measure all three conditions on one sensor, trying to equilibrate the F4-TPP⁺ concentrations between measurements? 2. The authors' interpretation of the negative charge translocation should be explained. EmrE was shown to transport in multiple stoichiometries and directions. Maybe this explains why the charge translocation becomes negative under these conditions? 3. Some representative current traces would further help understand

these results. The charge translocation is only a fraction of the information which can be derived from that kind of measurements. From the time-resolved current traces, it may become visible that the negative charge translocation has a different origin than the positive charge translocations.

Since the relevance of the authors' manuscript clearly comes from the structural data and the authors' interpretation of the functional data is fundamentally correct, I would highly recommend the manuscript for publication in Nature Communications.

A slight revision of the description for the functional SSM-based electrophysiology assay would be beneficial for the understanding of this section and for being able to reproduce the work.

Andre Bazzone

REVIEWER COMMENTS

Reviewer #1 (Remarks to the Author):

This manuscript focuses primarily on the NMR structure of EmrE in a lipid bilayer that has a bound drug. This NMR work is solid but the details of computational work and approach could be improved.

General Comments:

1. Why choose DMPC for NMR work?: Since this protein is a bacterial transporter in *E. coli*, why use a non-native lipid, DMPC. PE would potentially be a more realistic choice as bacterial membranes are roughly 70-80% PE. Some discussion on this and comparison to past NMR work in various lipids (done by the corresponding author) should be clearly stated.

This is a common question given the growing appreciation for the impact of lipid environment on integral membrane protein function. There is some data suggesting that EmrE function may be lipid dependent (see the work of Paula Booth about 10 years ago). However, we have a paper currently being prepared for submission that shows that changing lipid composition (acyl chain unsaturation, PC vs PE vs PG headgroups) has no significant effect on either the affinity of TPP+ for EmrE OR the rate of alternating access in the TPP+-bound transporter. Thus, lipid composition does not appear to have a significant effect on the drug-bound state and any lipid-dependent effects must occur on the more dynamic drug-free transporter. Since we are examining the drug-bound transporter here, the choice of lipid is not critical and we have used DMPC for simplicity and because many of the prior NMR studies of EmrE have been carried out using this lipid.

2. Are the MD Simulations long enough?: The authors have some detail of the stability of the structure (or lack of) in the supporting materials (Figure S9). The authors claim that Figure S9A is RMSF but this is not typically how RMSF is reported. Usually root-mean-squared fluctuations are reported based on a per residue basis (showing residue fluctuations about a STABLE average). I suspect this is the root-mean-squared deviation (RMSD). Either way, the structure has NOT stabilized. There is still a visible drift in increasing RMSD for the last 100ns. The authors must continue these simulations longer until there is at least 100ns of stable RMSD.

We thank the reviewer for pointing out these simulation details. We have updated the manuscript to state that the equilibration of the MD trajectory was based on RMSD rather than RMSF; Figure S10 is revised to reflect this point. We agree that the original MD simulation trajectory had not quite reached a complete plateau. However, long timescale MD trajectories are known to sometimes sample non-native, off-pathway states that may require simulations to several microseconds to return to native states (Heo and Feig, PNAS 2018, 13276-13281). To avoid this potential problem, we decided to keep the MD stage not overly long, with an operating definition of equilibration being when the RMSD time course reaches >95% of the final value when the RMSD rise is fit to biexponential growth. Since we also revised our simulations slightly to address reviewer #3's comments, we now show updated RMSD plots in Fig. S10 along with the biexponential fit. Analysis of this biexponential fit shows that at 250 ns, the RMSD is at 99.6% and 99.8% of the extrapolated 10 μ s value in the two runs. Therefore, we believe that these simulations are sufficiently equilibrated. The revised structural ensemble is now shown in Figure S10.

3. No discussion of NMR vs. MD structure differences: There lacks a discussion on why MD shows a more compact binding pocket compared to the NMR structure. There are other significant deviations from NMR. What if the authors started a second simulation imposing a structure similar to NMR? Would the structure be stable or revert to the current MD simulations based on the low-resolution x-ray structure? Currently this paper focuses strongly on NMR and lacks any discussion of the simulation work in the Discussion section and any insight in why there are differences.

The aim of this study is to obtain direct experimental constraints of the structure of the substrate-binding pocket of EmrE, rather than studying which aspects of previous literature MD simulations may be improved to obtain better agreement with experiments. These prior MD models had been reported by two of the most reputable labs in biomolecular simulations (Karplus and Tajkhorshid). But they were simulated without any experimental protein-drug contacts. Thus, we believe the only way to make further progress in understanding EmrE's substrate-bound structure is to carry out real measurements. It is outside the scope of this paper to investigate whether a different structural model might be unstable in MD simulations. Moreover, the previous X-ray crystal structure of EmrE has much lower resolution than the current NMR structure, which is constrained by 466 torsion angles (ϕ , ψ , χ_1) and 213 protein-drug distances. Thus, differences between our NMR refined structural model and the previous MD models are mainly due to these experimental constraints. We have now revised our discussions of the differences between the experimental structural model and the previous MD model for clarity.

4. Proper citation of force fields: The authors only cite the protein force field, when lipids and water are used. The CHARMM36 lipid force field should be cited as well as TIP3P.

Thank you for pointing out the missing references. We have now added citations to the original TIP3P paper (Jorgensen, Chandrasekhar, and Madura, *J. Chem. Phys.*, 1983) and its modification for CHARMM force field (MacKerell, ..., Karplus, *J. Phys. Chem. B*, 1998). We have also added the citation for the CHARMM36 lipid force field (Klauda, ..., Pastor, *J. Phys. Chem. B*, 2010). These references are added under the 'Structure Calculation of F₄-TPP⁺-bound EmrE' section in the Methods section.

Reviewer #2 (Remarks to the Author):

See attached file for more clarity

The SMR transporters display a broad specificity and provide resistance against various toxic compounds by active removal in an H⁺-coupled mechanism. The study described in the paper by Shcherbakov et al. provides important and long-awaited structural information to understand the substrate promiscuity of this vital family of transporters. This paper presents interesting information, but, many aspects need thorough revision. The claim that they report a 2.2 Å structure using ssNMR is very misleading. At most, using ssNMR they identified some amino acids interacting with the substrate, they determined several distances, and they use MD to model a structure. The resolution they are talking about is a mean pairwise RMSD between the structures. I would call it what it is, not a resolution. The abstract is grossly misleading and should be worded appropriately.

In this study we measured 213 distances between the drug and 72 residues of the protein to constrain the structure of the drug-binding pocket. Thus, the number of residues we have information for is not small. Together with 466 (ϕ , ψ , χ_1) torsion angles, the structure of the binding-site is one of the best constrained by NMR for a protein-substrate complex. The protein heavy-atom RMSD of 2.1 Å is defined as pairwise RMSD among 10 lowest-violation structures. This is given in Table 1. To our knowledge, it is accepted terminology to call this RMSD for NMR structures as the "xxx Å structure" of a protein. Since the definition of RMSD is spelled out fully in the paper, we believe it is more practical (and also due to space limitation) to adopt the simple language of "2.1 Å structure" in the abstract. Otherwise, we would have to write a long sentence such as "we report an experimental structure of drug-bound EmrE in phospholipid bilayers, determined using ¹⁹F and ¹H solid-state NMR and a fluorinated substrate tetra(4-fluorophenyl) phosphonium (F₄-TPP⁺), which has a pairwise heavy-atom RMSD of 2.1 Å in the lowest-violation ensemble".

The introduction is appropriate for a grant request or a review. The first paragraph is a gross overstatement of the importance of EmrE in antibiotic resistance. As far as I know, EmrE is an interesting model system for structure-function studies, and it transports toxic compounds. It confers resistance to them, but it is a minor player, if at all, in antibiotic resistance. This paragraph should undoubtedly be toned down.

The second paragraph is OK for a review but not relevant for this paper. In this work, residues interacting with

a substrate were identified. How can they conclude anything about symport or antiport? High-resolution structural information of the substrate-binding site is essential for understanding promiscuity of substrate recognition, but they say very little about the possible coupling mechanisms.

We have thoughtfully considered the reviewers concerns and have rewritten the first two paragraphs of the intro to make clear what is known and not known about EmrE specifically, and to more explicitly state why the information presented in these paragraphs is relevant in driving us to undertake the particular experiments presented here.

In vitro EmrE has all the hallmarks of antibiotic resistance; clinical presentation of antibiotic resistance requires other proteins. The reviewer is correct that simple gene knockouts suggest a more significant role in clinically relevant drug resistance for AcrABToIC in *E. coli*. Because of the broad and overlapping specificity of MDR transporters, individual gene knockouts rarely unmask a strong drug susceptibility phenotype. EmrE is located in the inner membrane and pumps substrates out of the cytoplasm or into the periplasm. AcrA/B/ToIC or OmpW (see Beketskaia 2014 in *J. Bacteriology*) are required as partners to move substrates pumped out of the cytoplasm by EmrE, further complicating traditional genetic knockout approaches to determine EmrE activity *in vivo*. However, the easy selection of antibiotic resistance for erythromycin and other clinical antibiotics in the lab, including drugs used to treat UTI (of which *E. coli* is a major cause) suggests that clinically relevant antibiotic resistance may arise. There is clear data demonstrating that EmrE contributes to bacterial survival under conditions of osmotic shock and pH stress, confers resistance to bile salts and other host molecules, and contributes to *E. coli* virulence.

The introductory description that EmrE can both symport and antiport substrates and protons is relevant because our long-term goal is to understand how different EmrE substrates trigger different types of proton-coupled transport. We hypothesize that difference in how substrate interacts with the transporter underly these distinct phenotypes, but further exploration of that question requires the ability to obtain precise experimental data to localize the substrate within the binding pocket, which is what we present in this manuscript.

Some specific points, some of them minor:

We used solid-supported membrane electrophysiology to monitor liposomal transport of F₄-TPP⁺ by WT-EmrE. These results demonstrate that F₄-TPP⁺ binds and is antiported by EmrE in a similar manner to other EmrE substrates.

I am not sure whether the use of this method was described before. If it was, it should be cited. If this is the first time to be used with EmrE substrates, more information should be provided, at least as a supplementary figure. The authors state F₄-TPP binds and is transported similarly to other substrates. How are other substrates transported when this technique is used? Where can we find this information?

Solid-supported membrane electrophysiology is a newer technique for monitoring liposomal transport. It was developed to enable the use of electrophysiology to monitor the much slower flux through transporters (relative to ion channels) by Klaus Fendler. It has been used to study transport by a number of other transporters now that it is commercially available (see <https://www.nanion.de/en/products/surfe2r-n1.html> for some examples using the instrument we use in our lab). It has been used to study the SMR transporter Gdx (application note from Randy Stockbridge available at the above website, manuscript still under review). We have a preprint available on BioRxiv describing the development of an SSM-based method to determine transport stoichiometry quantitatively that presents additional data on Gdx transport to establish the use of this method in our hands (<https://www.biorxiv.org/content/10.1101/2020.05.07.082438v1.full>). This manuscript is being revised for submission to a different journal. Because the EmrE transport mechanism is complicated and transport is not stoichiometric in the traditional sense, a discussion of different substrate transport by SSM is well beyond the scope of this manuscript. The data and controls presented here show that EmrE antiports F₄-TPP⁺, which is similar to the antiport of other substrates previously reported by numerous labs using radioactive transport assays.

Besides, I am surprised to see the results. If I remember correctly, the transport of TPP is extremely slow. Is the rate high enough to give an electrical signal?

The reviewer is correct that TPP⁺ flux is slow (see Robinson *et al* 2017 *PNAS* 47:E10083-E10091 for direct measurement of the counter-transported protons during TPP transport, and Robinson *et al* 2018 *Anal. Biochem.* 549:130-135 for readout of TPP transport into liposomes, both of which show transport occurs on the second timescale), but not so slow that it cannot be recorded with this method (see the new SI Figure S3 to see the raw electrical signal recorded).

Is the rate of transport higher than the passive permeabilities of other ions?

Given the slow rate of transport, this was a potential concern before the experiments were performed. However, since we observe minimal flux in proteoliposomes reconstituted with transport-dead EmrE, the origin of the signal we detect cannot be passive ion. Passive ion flux would occur equally in EmrE and E14Q-EmrE proteoliposomes. Thus, this is ruled out by the control experiments presented in Figure 1 in the paper.

They show the integrated current and used an inactive mutant as control. Did they try liposomes without any protein? They should show the actual time course of the response as well.

We thank the reviewer for pointing out that we failed to show the raw current data. We have now added SI Fig. S3, which shows this data. We did not include lipid-only liposomes in this set of experiments. We did include those in our original experiments (see BioRxiv preprint at <https://www.biorxiv.org/content/10.1101/2020.05.07.082438v1.full>). The results for lipid-only liposomes and E13Q-Gdx (transport-dead) proteoliposomes were nearly identical. We did not repeat those experiments here because the E14Q-EmrE proteoliposomes are a better control. Insertion of protein into liposomes can introduce small membrane defects that increase passive ion permeability (particularly for protons, which are very small). Thus, comparing proteoliposomes containing the same amount of EmrE with the only difference being whether EmrE is functional or non-functional provides the best control to isolate ion flux due to transport from any background signal due to buffer exchange and passive ion permeability.

Cross-peak intensity buildup (Fig. 2E) indicates a time constant of 16±2 ms for the exchange, meaning that F₄-TPP⁺ reorients or undergoes tetrahedral jumps in the binding pocket with a rate of ~50 s⁻¹ at ambient temperature.

Since the results described here are possible because of the slow-changing S64V-EmrE they should compare the above rates to those observed in the wild type.

We agree that comparing the drug dynamics in the mutant and WT binding pocket will be very interesting and we have plan to do in the future. But it is outside the scope of the current manuscript as producing isotopically labeled WT protein and studying its binding to F₄-TPP⁺ involves significant additional work in a new direction. The current paper focuses on determining the structure of the complex using the less dynamic S64V-EmrE mutant. The reason to present these drug dynamics data in this paper that it shows the heterogeneous environment of the drug due to contact with different protein sidechains, and the drug dynamics at high temperature implies a relatively spacious binding pocket, which is consistent with our distance data obtained at low temperature. Hence the drug dynamics is well connected with the protein structure.

The pH of the final samples was adjusted to 5.8 at 45oC using a Hamilton biotrode microelectrode. The sample was then incubated with excess solid F₄- TPP⁺ at 45oC overnight. Excess F₄-TPP⁺ was removed using microcentrifugation prior to transferring solution to NMR tubes.

The affinity of TPP at this pH is very low, and a significant fraction is released in the NMR tubes, as shown by the results in figure 2B. Of the five components, only one (110 ppm) arises from fluorines close to the protein. I understand the 105.7 peak attributed to free TPP, but what is the meaning of the -106 peak: this fluorine is the furthest from the protein? Is it bound? Is it free?

The sample contained a modest excess of F₄-TPP⁺ relative to the protein, in order to ensure saturation of the binding pocket. The observed REDOR dephasing to an S/S₀ value of ~0 for some sites such as W63Ae indicate that all proteins are saturated. Thus, the reduced ¹³C-¹⁹F CP transfer to the -106 ppm peak suggests that this peak has partial contributions from lipid-bound TPP⁺. This is consistent with the data by Glaubitz and coworkers (Ong et al, JACS, 2013), which also detected non-specific lipid-bound TPP⁺ in addition to protein-bound TPP⁺. This is not surprising given the hydrophobic nature of TPP⁺. We have now clarified this result description and added the citation to the 2013 paper.

Do they have an estimate of activity after such a prolonged incubation at the low pH and high temperature??

We recognize the reviewers' concern and would like to make very clear that we prepare the solution NMR samples in bicelles AND the solid-state NMR samples in liposomes AND the transport assay proteoliposomes in an identical manner. We prepare our bicelles by first making proteoliposomes (Stop here and pack in rotors for SSNMR! Stop here and extrude for transport assays!) and then adding short-chain lipid to break the proteoliposomes up into bicelles. This results in the most stable samples for any of our biophysical or biochemical experiments. We have obtained solution NMR spectra of the F4+-TPP bound protein immediately after preparation and after several weeks (after ZZ-exchange experiments at pH 5.8). The 2D spectra are identical. Further, the hNH NMR spectrum in the solid-state remains identical over a year after initial measurements. EmrE is an incredibly stable protein, and we have acquired 2D NMR spectra at pH values ranging from 4 – 11 (see Morrison *et al* 2015 *J Gen Physiol* 146:445-461, and Robinson *et al* 2017 *PNAS* 47: E10083-E1009). In our prior work there was no change in peak position or intensity at pH values between 5 and 8.5, at more extreme pH values we did see some spectral changes over long times (days), which we attribute to acid and base-catalyzed lipid hydrolysis (monitored by TLC) and the consequent effect of slowly converting the bicelles to lysolipid micelles. NMR is extremely sensitive to any change in protein structure or dynamics, so the stability of the spectrum over weeks and months provides an excellent indicator that the protein remains in a stable state.

From the abstract:

EmrE forms an antiparallel asymmetric dimer that causes multidrug resistance in bacteria by effluxing toxic polyaromatic cationic drugs in a proton-coupled manner.

The dimer causes? Please reword.

We have reworded the abstract to clarify our intended point that it is the dimer that is the functional unit, which confers multidrug resistance. We thank the reviewer for pointing out the confusion caused by the original wording.

Reviewer #3 (Remarks to the Author):

When it was first described by Shimon Schuldiner, EmrE has been considered as the ideal model system for secondary transporters and multidrug efflux due to its small size and apparent simplicity. Ironically, this relatively small membrane protein with just four transmembrane helices turned out to be a complex experimental challenge. It forms an antiparallel homodimer and was the first dual topology transporter described. Extensive efforts to obtain a high-resolution structure - including NMR, Xray and cryo-EM - failed so far but at least all data emerged into a consensus model. However, the lack of real high-resolution data

still prevents deriving a good mechanistic model. One key problem has been the unfavorable intrinsic dynamics of EmrE which might have prevented progress in structure determination so far.

In this paper, Hong and Henzler-Wildman present the first high resolution structure of EmrE. The work presents excellent and exciting progress in the field and was only possible due to the combination of four experimental tricks/methods: The structure determination is based on solid-state NMR, which enables to work directly in native-like bilayers. Especially noteworthy is the fact that a large number of valuable long-range contacts could be obtained through the use of ^1H - ^{19}F REDOR experiments. For this purpose, the high affinity substrate TPP was fluorinated. Partial assignment was achieved by ^1H and ^{13}C detected 3D experiments. However, most importantly for this work has been the introduction of the S64V mutation, which alters the protein dynamics in such a way, that a great spectral resolution could be achieved. All previous solid-state NMR studies on the EmrE wild type suffered from line broadening due to an unfavorable motional regime.

In my opinion, this work is a great step forward. Data quality is excellent and data interpretation is sound, but I have some minor comments / suggestions / questions which might help improving the paper:

- For me it is still puzzling why/how a homodimer in complex with a symmetric substrate assumes an asymmetric structure and even shows 'asymmetric dynamics' as highlighted in the manuscript with A being in stronger complex with the substrate compared to B. What determines the different behavior of A and B?

We recognize that this point has been confusing to many scientists who are not intimately familiar with EmrE. The answer lies in the antiparallel orientation of the two subunits of the homodimer. Transporters are only open on one side of the membrane at a time, this means that in an antiparallel homodimer one subunit has the N- and C-terminal tails packed against the other subunit closing off access, while the N- and C-terminal tails of the other subunit are on the open side exposed to water and not in contact with the other subunit. You can think about inverting one of your hands relative to the other to act as the antiparallel subunits and then bringing the fingers of one hand to the palm of the other to close off one face of the transporter. Your fingertips and palms are now in different environments. Thus, as soon as you have antiparallel homodimer closed on one side, as required for active transport, this one-sided closure introduces an asymmetry between the subunits. In EmrE, all of the cryoEM and x-ray structures clearly show asymmetry between the two subunits of the homodimer. This results in different conformations of the two subunits, different chemical shifts of the residues within each subunit, different pKa values for the two E14 residues, etc. When substrate interacts with this asymmetric dimer, it preferentially binds to E14 on one subunit, presumably because this E14 is more likely to be deprotonated or because surrounding residues create a more favorable binding pocket for TPP+ or other substrates due to the particular conformation of that subunit.

- E14 plays an important role in the asymmetry argument and different protonation states in the substrate bound state could certainly explain the observations. Is it correct that only CD of E14A has been assigned (Tab. S1)? How about the different distances between TPP and E14A-CD and E14B-CD discussed in the paper? Are these experimental results or conclusions from the structure calculation?

The reviewer is correct that only one of the two E14 CD atoms is assigned in this paper. The distances discussed are indeed from the structural calculation, not directly measured. We thank the reviewer for pointing out this ambiguity, and have revised the Table 2 caption to clarify this point by stating "Protein-substrate distances extracted from the NMR-refined structural models", and clarified this in the Results section where the table is referenced (Results, structure section).

- Substrate dynamics, heterogeneity, TPP-E14 interaction: It might be helpful to refer to Ong et al. JACS 2013 (incl. SI) in which dynamics and heterogeneity of TPP and MTP within the EmrE pocket has been addressed by ^{31}P -MAS NMR and in which ^{13}C -TPP has been used for probing interactions with E14 by DNP. Overall, these results seem to fit with the data presented here. Regarding the ^{19}F line shape analysis

in Fig. 2A/B/C: Could some of the spectral components arise from non-specifically bound TPP populations as shown by Ong et al.?

We thank the reviewer for reminding us to make an explicit comparison of our data to the previous rigorous work of Ong *et al.* We have now added this citation and discussed the non-specific lipid-bound TPP⁺. Since some of the protein sites show complete REDOR dephasing to S/S₀ of 0, all protein binding sites are saturated. Thus, the small amount of lipid-bound TPP⁺ results from the excess drug that was titrated into the sample to ensure saturation. While it is in principle possible, all measured REDOR distance restraints are consistent with a single binding site near E14. We do not see any indication of a second binding site. Thus, we focus on that binding site in our structural analysis, which is the primary focus of the manuscript.

- Structure calculations: How unambiguous is the structure calculation? The reason for the question is that prior structural knowledge is needed for the substrate docking step and for assigning the F-H restraints. Are there any controls which could be done?

This is a good question, and we have now revised our structure calculation to address this issue of uniqueness of the structure. Two MD simulated models have been reported for EmrE – one from the Karplus group and the other from the Tajkhorshid group (both in 2018). In the original submission, we only considered the Karplus apo model for docking and structure calculation. To investigate whether a different starting model might result in a different NMR structure that is equally consistent with the experimental constraints, we have now additionally docked the substrate into the Tajkhorshid model. Importantly, we found that this Tajkhorshid model resulted in 4 different binding sites, only one of which was at the dimer interface. Because of this poor convergence, we did not continue to calculate the structure using these docking results. We have now added the Tajkhorshid model's docking result in the revised Figure S10, panel B.

Also, in contrast to NOE or PDSD restraints, REDOR distances are precise measurements. What determined the upper and lower limits for the structure calculation?

Indeed, REDOR measurements are quantitative, thus the distance uncertainty was extracted from fitting the H-F REDOR decays shown in Fig. S8. The best-fit distance was obtained from the minimum RMSD between the experimental and simulated REDOR curves. The uncertainty was obtained from an RMSD threshold value (blue horizontal line): any distances with RMSD below the threshold are considered to agree with the data. We chose the threshold as the maximum scatter for sites that do not dephase. For example, G97B does not display dipolar dephasing, but the S/S₀ ranges from ~0.83-1.03. This is now clarified in the methods section.

How important are the phi/psi angles used here and is it a problem that not all residues have been assigned? Could the discrepancy between A and B (72 vs. 54 residues assigned) affect the structure calculation in terms of asymmetry?

We do not believe that incomplete chemical shift assignment affects the structure calculation, because the unassigned residues are predominantly located in interhelical loops, which are far from the drug binding site. To further reduce the chance of the effect of incomplete torsion angles on the structure calculation, we re-ran the MD simulations using a more extended list of TALOS (ϕ , ψ) torsion angles and sidechain χ_1 angles obtained from solution NMR chemical shifts. This is valid because the solid-state and solution NMR chemical shifts agree very well (Fig. S8), indicating that the protein conformations are indistinguishable in DMPC bilayers and DMPC/DHPC bicelles. These torsion angles are inputted in both the HADDOCK and GROMACS stages.

Knowing more backbone torsion angles for monomer A than monomer B is very unlikely to cause any artificial asymmetry in the calculated structure, because the EmrE asymmetry lies more at the tertiary structure level, in terms of different interhelical packing and helical tilt angles between monomer A and B.

Except for TM3 helix, the chemical shift differences between monomer A and monomer B are small, as shown in Figure S6.

Reviewer #4 (Remarks to the Author):

Multidrug resistance (MDR) is one of the greatest threats to human health. In contrast to other transporters, MDR transporters have a rather complicated mechanism due to their broad substrate specificity. EmrE is a well-studied model protein for MDR. There is evidence that EmrE is able to perform antiport, symport, and uniport under different conditions, which further enhances the complexity of its transport mechanism. This is a rather new field as transporters have been believed to be strictly coupled; By theory, antiport and symport contradict each other. For these reasons, a deeper insight into the transport mechanism of EmrE is of broad interest to the scientific community.

Structural data on EmrE is already available and helped to understand its functional asymmetry and dual topology. But the authors report the first high-resolution structure of drug-bound EmrE, which allows for important conclusions about drug-transporter interactions on the side chain level. In addition, the authors' report includes the first observation of substrate dynamics at the binding pocket. This and other mechanistic insights also help developing inhibitors to face MDR.

The authors used novel approaches to counter the issues responsible for the lack of high-resolution structural data on MDR transporters: (1) They developed a mutant (S64V-EmrE) with similar drug-binding properties, but slower conformational dynamics. (2) They applied a newly developed long-range ¹H-¹⁹F distance NMR technique. (3) And they also found a drug (F4-TPP+) with reduced conformational dynamics.

To validate that F4-TPP+ is indeed a transported EmrE substrate, the authors applied SSM-based electrophysiology – a rather novel approach for the measurement of transport activity in membrane vesicles or proteoliposomes.

The authors used pH jumps to activate EmrE, which is a difficult approach due to possible solution exchange artifacts. The authors were aware of that, designed a well-suited solution exchange protocol, and used proper control experiments: The same measurements using the transport deficient E14Q EmrE mutant did not generate significant signals on the SSM, proving that the currents detected for WT-EmrE are due to proton-coupled F4-TPP+ transport by EmrE.

Another difficulty is the establishment of the chemical gradient for F4-TPP+ during the functional assay. The description in the method section indicates that the sensors were equilibrated with the respective F4-TPP+ concentration prior to the measurement. From that description, it is not clear if they equilibrated by solution exchange using the measurement device or during sample preparation and before sonication of the sample. When F4-TPP+ was equilibrated during sonication, the concentrations inside and outside the vesicles should perfectly equilibrate. But when equilibration was performed during solution exchange, the concentrations inside the vesicles may not equilibrate to the exact concentrations of the buffer due to the lipid barrier and rise in membrane potential when F4-TPP+ crosses that barrier. The membrane potential serves as a counterforce which may prevent further F4-TPP+ equilibration, even when longer incubation times are used.

The reviewer is correct that this is a potential concern, as we did perform the equilibrations on the instrument. Equilibrations were performed using multiple washes of internal buffer, while recording any changes in signal. Washes were performed until successive washes produced no observable current. I have added this detail to the method section of the manuscript. We have data that demonstrates that this method can reliably and accurately exchange the internal buffer contents (in a preprint publication available on BioRxiv at <https://www.biorxiv.org/content/10.1101/2020.05.07.082438v1.full>), but as the reviewer notes in the next paragraph, quantitatively accurate concentrations are not necessary for qualitative assessment of whether F4-TPP+ is transported, which is what we claim here.

However, the analysis used by the authors shows that EmrE transport activity is enhanced, when the direction of the F4-TPP⁺ gradient is in the opposite direction than the pH gradient. This makes perfect sense for an antiport mechanism. The conclusion is also perfectly valid, even when the exact concentrations inside the vesicles are not known.

For proton and F4-TPP⁺ gradients in opposite directions, a large positive charge is translocated, showing that more protons are imported than F4-TPP⁺ molecules are exported per transport cycle. When only the proton gradient serves as a driving force, the translocated charge decreases, rendering one step in F4-TPP⁺ translocation as the rate-limiting step; And showing that the authors really observe F4-TPP⁺ translocation. However, when both cationic gradients are inward-facing: Why does the translocated charge turn negative, when the transporter is activated by an increase in external proton concentration? To my understanding, this is only possible, when both gradients are established at the time to drive transport, e.g. a simultaneous F4-TPP⁺ and pH jump; or when the pH gradient is established before activation of EmrE via F4-TPP⁺ concentration jump; In these experimental settings, the result would indicate that the inward-facing F4-TPP⁺ gradient drives efflux of protons, even against the proton gradient.

The reviewer is correct. We have revised the methods section to make clear that the gradients are instituted simultaneously.

But in the methods section, the authors state that F4-TPP⁺ concentration is equilibrated before the measurement which is triggered by pH jump. The increase of proton concentration on the external side can only trigger proton influx; The translocated charge has to be positive and the minimum charge translocation observed (for zero transport activity) should be zero. Further clarification is required here.

We thank the reviewer for pointing out this point of confusion. In our experiments, only the internal drug concentration is set prior to the pH jump. The methods section has been clarified to indicate that the drug and proton gradients are set by a simultaneous drug and proton jump.

Taken together, the understanding of the SSM-based electrophysiology assay would be enhanced when the following points are considered: 1. More details about the experimental settings should be given, e.g. details about pre-loading the vesicles with F4-TPP⁺, incubation times between measurements, and the time points of when the pH and F4-TPP⁺ concentrations are changed at the sensor. Are all three conditions shown measured on different sensors which have been equilibrated with the given F4-TPP⁺ concentration? Or did the authors measure all three conditions on one sensor, trying to equilibrate the F4-TPP⁺ concentrations between measurements?

We thank the reviewer for pointing out the need for additional experimental details. We have revised the methods section to clarify each of these points. The only thing not explicitly in the methods is the incubation time, as no incubation time was used apart from the time required to acquire measurements.

2. The authors' interpretation of the negative charge translocation should be explained. EmrE was shown to transport in multiple stoichiometries and directions. Maybe this explains why the charge translocation becomes negative under these conditions?

We thank the reviewer for pointing out the confusion caused by original wording. We have revised the main text to clarify that the negative charge movement results from the reversal of driven transport direction.

3. Some representative current traces would further help understand these results. The charge translocation is only a fraction of the information which can be derived from that kind of measurements. From the time-resolved current traces, it may become visible that the negative charge translocation has a different origin than the positive charge translocations.

We thank the reviewer (and reviewer 2) for pointing out this omission. We have added SI Fig. S3 with the time-resolved current traces. There are multiple components to the signal, regardless of the direction of

transport. There is not enough data here to definitively interpret these components separately. We are working on a separate manuscript to more thoroughly analyze the multiple components observed in the time-resolved current traces for EmrE transport of a number of different substrates. The complexity of the EmrE transport cycle renders this more detailed analysis beyond the scope of the current manuscript, which is focused on the structure of substrate-bound EmrE. The integrated current reflects the total net charge flux, providing a first-order analysis of which direction net charge moves in response to a particular gradient condition and that is sufficient to demonstrate active transport.

Since the relevance of the authors' manuscript clearly comes from the structural data and the authors' interpretation of the functional data is fundamentally correct, I would highly recommend the manuscript for publication in Nature Communications.

A slight revision of the description for the functional SSM-based electrophysiology assay would be beneficial for the understanding of this section and for being able to reproduce the work.

Andre Bazzone

** See Nature Research's author and referees' website at www.nature.com/authors for information about policies, services and author benefits.

REVIEWER COMMENTS

Reviewer #1 (Remarks to the Author):

Revised manuscript is acceptable.

Reviewer #2 (Remarks to the Author):

The authors have addressed some of the points I raised but not in an entirely satisfactory way.

I have no problem accepting the contention that F-TPP is a substrate of EmrE. This conclusion is evident from the structure that reveals a molecule of F-TPP bound and the compound's similarity to TPP. However, the authors tend to make sweeping conclusions based on insufficient data. I am quite surprised by the quality of the SSM experiments' raw data, data that the authors chose not to show in the first version of the article. The data is in stark contrast with the clear, straightforward, and elegant results using the Guanidinium transporter in their Biorxiv paper and those of Stockbridge et al. I understand that TPP is not guanidinium. Still, then one should be more careful in the interpretation. The figure legend is entirely accurate: "... It is possible that the initial spike represents substrate binding or turnover of proton-bound EmrE, while the slower process likely represents alternating-access conformational exchange and transport. "Yes, it is possible, but either you properly address the issue or you do not conclude: " The data and controls presented here show that EmrE antiports F4-TPP+, which is similar to the antiport of other substrates previously reported by numerous labs using radioactive transport assays." The data is not similar; the controls are not enough.

The authors' tendency to "simplify" the message is exemplified by their response to their claim of a "2.2 Å structure". I am not an NMR person, and, as such, I understand resolution more strictly and, alas, in science, one should try to be as accurate as possible so that the audience is not misled. I have looked in the PDB, and none of the NMR structures claims for resolution. I have randomly looked at the abstracts of 10 papers that describe structures obtained by solution and ssNMR techniques, and none of them talks about an xx structure. There is nothing wrong writing what has been done: "we report an experimental structure of drug-bound EmrE in phospholipid bilayers, determined using ^{19}F and ^1H solid-state NMR and a fluorinated substrate tetra(4-fluorophenyl) phosphonium (F4-TPP+), which has a pairwise heavy-atom RMSD of 2.1 Å in the lowest-violation ensemble" or something similar. I trust the authors that they can also shorten this sentence.

Last but not least, their claim that they keep repeating in every paper that EmrE functions as both antiporter and symporter. Maybe I am not fully updated with the literature. I tried to find evidence about symport, and I read the papers 15 and 17 referenced by the authors. In reference 15 they describe a model that, like every model, needs to be experimentally validated before it is accepted. In reference 17, they claim that their model "... model accounts for previously inexplicable behaviors of EmrE, such as the ease of converting the protein to a symporter."The protein may be converted to a symporter by mutation. I could not find evidence that the wild type protein is a symporter as well as an antiporter. If there is such evidence that I may have missed, they should cite it properly.

Reviewer #3 (Remarks to the Author):

The authors have addressed all of my concerns.

Reviewer #4 (Remarks to the Author):

The authors have addressed all my concerns and improved the manuscript accordingly. The method section now allows the readers to easily follow and understand the experimental setup for the electrophysiological experiments.

Thank you also for adding Figure S3. This makes the origin of the integrated charges shown in Figure 1 very clear. Given the authors explanations, the interpretation for the biphasic current shape is sufficiently clear to allow for the analysis and conclusions the authors present in Figure 1. There is a fast pre steady-state current very likely representing drug binding which is not affected by the gradients. The slow current component on the other hand represents transport. It is affected by the substrate gradients as expected. This also becomes clear by comparison with the negative control sensors (transport dead mutant), which only show a small and fast current, lacking the slow transport component completely.

Reviewer #2 (Remarks to the Author):

The authors have addressed some of the points I raised but not in an entirely satisfactory way.

I have no problem accepting the contention that F-TPP is a substrate of EmrE. This conclusion is evident from the structure that reveals a molecule of F-TPP bound and the compound's similarity to TPP. However, the authors tend to make sweeping conclusions based on insufficient data. I am quite surprised by the quality of the SSM experiments' raw data, data that the authors chose not to show in the first version of the article. The data is in stark contrast with the clear, straightforward, and elegant results using the Guanidinium transporter in their BioRxiv paper and those of Stockbridge et al. I understand that TPP is not guanidinium. Still, then one should be more careful in the interpretation. The figure legend is entirely accurate: "... It is possible that the initial spike represents substrate binding or turnover of proton-bound EmrE, while the slower process likely represents alternating-access conformational exchange and transport. "Yes, it is possible, but either you properly address the issue or you do not conclude: " The data and controls presented here show that EmrE antiports F4-TPP+, which is similar to the antiport of other substrates previously reported by numerous labs using radioactive transport assays." The data is not similar; the controls are not enough.

We thank the reviewer for pointing out that this was still not entirely clear and have revised the manuscript in the main text to make explicit that the reversal of current is what is most critical for identifying coupled antiport and is what is similar between the Gdx data. We also revised the Fig. S3 legend to more clearly lay out the context for the interpretation of the data.

The authors' tendency to "simplify" the message is exemplified by their response to their claim of a "2.2 Å structure". I am not an NMR person, and, as such, I understand resolution more strictly and, alas, in science, one should try to be as accurate as possible so that the audience is not misled. I have looked in the PDB, and none of the NMR structures claims for resolution. I have randomly looked at the abstracts of 10 papers that describe structures obtained by solution and ssNMR techniques, and none of them talks about an xx structure. There is nothing wrong writing what has been done: "we report an experimental structure of drug-bound EmrE in phospholipid bilayers, determined using ^{19}F and ^1H solid-state NMR and a fluorinated substrate tetra(4-fluorophenyl) phosphonium (F4-TPP+), which has a pairwise heavy-atom RMSD of 2.1 Å in the lowest-violation ensemble" or something similar. I trust the authors that they can also shorten this sentence.

We have now removed the structure resolution information from the abstract. This information is given in the main text and does not need to be stated in detail in the abstract.

Last but not least, their claim that they keep repeating in every paper that EmrE functions as both antiporter and symporter. Maybe I am not fully updated with the literature. I tried to find evidence about symport, and I read the papers 15 and 17 referenced by the authors. In reference 15 they describe a model that, like every model, needs to be experimentally validated before it is accepted. In reference 17, they claim that their model "... model accounts for previously inexplicable behaviors of EmrE, such as the ease of converting the protein to a symporter." The protein may be converted to a symporter by mutation. I could not find evidence that the wild type protein is a symporter as well as an antiporter. If there is such evidence that I may have missed, they should cite it properly.

We respectfully disagree with the reviewer on this point. In Reference 17, we introduce the free exchange model for EmrE as the simplest model that includes all of the known states and transitions of EmrE observed by NMR – both by our group and the group of Nate Traaseth (NYU). We then performed liposomal transport assays (see Figure 7 in the main text of reference 17). These assays clearly show that EmrE is not a tightly coupled antiporter. The simplest explanation is that EmrE can also perform symport. We noted in that paper that the data could also be explained by a combination of coupled antiport and substrate-triggered uncoupled proton flux, with these transport pathways together resulting in net transport of both substrate and proton in the same direction across the membrane. In the intro for this manuscript we

were careful to accurately report both possibilities. We summarize the data from Ref 17 here to illustrate the experimental data underlying this statement:

- EmrE was reconstituted into proteoliposomes and then a transmembrane voltage was created using potassium/valinomycin. We directly monitored external pH to watch proton movement into or out of the liposome. No significant proton flux was observed upon creation of a transmembrane voltage (arrow marked V in Fig. 7F and 7G). This was separately assayed using a different readout with the same result (Fig. 7B). These assays clearly show that EmrE does not perform uncoupled proton flux, since a transmembrane voltage should cause proton flux if there is any leak pathway available.
- No proton flux occurs upon creation of a transmembrane voltage OR addition of substrate in proteoliposomes reconstituted with E14Q-EmrE (transport dead point mutant). This control further confirms that any proton flux observed is mediated by active EmrE transport.
- For EmrE proteoliposomes, addition of substrate to the outside of the liposomes in the absence of a transmembrane voltage triggers a small drop in pH. This is as expected for antiport (substrate added externally creates an inward concentration gradient that drives substrate in and proton out).
- A positive-inside membrane potential (Fig. 7F) results in a larger pH drop when substrate is added externally. This matches many prior experiments in the literature and again reflect 2:1 H⁺/ethidium⁺ antiport – as ethidium⁺ goes in and 2 protons come out there is net movement of +1 out, and this is favored by the positive-inside potential.
- A negative-inside membrane potential had never been tested previously (or at least not reported in the literature). If EmrE only performs 2H⁺/1 ethidium⁺ antiport, then a negative-inside potential should *inhibit* transport. Instead, we observed that protons moved in the opposite direction (Fig. 7G), *into* the liposome (external pH rises). Since ethidium⁺ is only added externally, it must move inward. We know from the controls described above that proton flux does not occur due to simple proton leak, and indeed in this assay there is no proton flux when the transmembrane voltage is imposed (arrow marked with V), only when the ethidium⁺ substrate is added (arrow marked with E). Thus, flux requires both substrate and proton have net movement into the liposome. Symport is defined by both substrates moving in the same direction across the membrane in a coupled manner (so that movement of one substrate down its gradient can drive movement of the other substrate against its gradient). We note that the experiments presented don't demonstrate robust energetic coupling between the substrates that move the same direction. Thus, it is also possible that substrate triggers a proton leak pathway through EmrE, and this combined with 2:1 antiport results in the net flux observed. However, the net inward movement of protons is exactly twice the magnitude when the voltage is negative inside compared to the outward movement of protons when the voltage is positive outside. This exactly matches the net +1 out for antiport and net +2 in for 1:1 symport by proton and ethidium⁺. This is what would be expected for coupled transport but is not sufficient in our opinion to definitively demonstrate symport by WT EmrE. Thus, we concluded that the simplest explanation for the data was symport, but also consider a combination of antiport and uniport possible under these conditions.

Consequently, Reference 17 does provide direct experimental evidence in support of our statement in the intro that “giving evidence that EmrE may function not only as a proton-coupled antiporter, pumping toxic polyaromatic cations out of *E. coli* (Fig. 1A), but also as a proton-coupled symporter or uncoupled uniporter^{15, 17}”. Reference 15 is a kinetic modeling paper that was published 2 years later and provides the theoretical foundation to explain how EmrE could exhibit both behaviors, since like the reviewer we at first found the idea that a single transporter could perform both symport and antiport difficult to comprehend. In that paper we also noted that this is not without precedent among other transporters.

Reviewer #4 (Remarks to the Author):

The authors have addressed all my concerns and improved the manuscript accordingly. The method section now allows the readers to easily follow and understand the experimental setup for the

electrophysiological experiments.

Thank you also for adding Figure S3. This makes the origin of the integrated charges shown in Figure 1 very clear. Given the authors explanations, the interpretation for the biphasic current shape is sufficiently clear to allow for the analysis and conclusions the authors present in Figure 1. There is a fast pre steady-state current very likely representing drug binding which is not affected by the gradients. The slow current component on the other hand represents transport. It is affected by the substrate gradients as expected. This also becomes clear by comparison with the negative control sensors (transport dead mutant), which only show a small and fast current, lacking the slow transport component completely.

We thank reviewer 4, who is one of the world experts in SSME technology, for the clear summary of the transport data.